# Electron cryo-tomography provides insight into procentriole architecture and assembly mechanism

Sam Li[1]*, Jose-Jesus Fernandez[2], Wallace F Marshall[1], David A Agard[1,3]*

[1]Department of Biochemistry and Biophysics, University of California, San Francisco, San Francisco, United States; [2]Centro Nacional de Biotecnologia (CSIC), Madrid, Spain; [3]Howard Hughes Medical Institute, University of California, San Francisco, San Francisco, United States

**Abstract** Centriole is an essential structure with multiple functions in cellular processes. Centriole biogenesis and homeostasis is tightly regulated. Using electron cryo-tomography (cryoET) we present the structure of procentrioles from *Chlamydomonas reinhardtii*. We identified a set of non-tubulin components attached to the triplet microtubule (MT), many are at the junctions of tubules likely to reinforce the triplet. We describe structure of the A-C linker that bridges neighboring triplets. The structure infers that POC1 is likely an integral component of A-C linker. Its conserved WD40 β-propeller domain provides attachment sites for other A-C linker components. The twist of A-C linker results in an iris diaphragm-like motion of the triplets in the longitudinal direction of procentriole. Finally, we identified two assembly intermediates at the growing ends of procentriole allowing us to propose a model for the procentriole assembly. Our results provide a comprehensive structural framework for understanding the molecular mechanisms underpinning procentriole biogenesis and assembly.
DOI: https://doi.org/10.7554/eLife.43434.001

*For correspondence:
samli@msg.ucsf.edu (SL);
agard@msg.ucsf.edu (DAA)

Competing interests: The authors declare that no competing interests exist.

## Introduction

The centriole is a barrel-shaped structure composed of a set of MT triplets arranged in a characteristic nine-fold symmetry. As an evolutionarily conserved organelle, the centriole, also known as the basal body, fulfills many cellular functions. In cycling cells, a pair of centrioles recruits pericentriolar material (PCM). Together they form the centrosome, the primary microtubule organizing center (MTOC) in animal cells. Related to its function as an MTOC, the centrosome is essential for mitotic spindle formation, spindle orientation and faithful mitotic chromosome segregation and for intracellular transport of cargoes. In non-dividing cells, the centriole functions as basal body to template cilium formation. Based on their structure and functions, the cilia can be further classified into primary or motile cilia. The primary cilium functions as a cell 'antenna' that senses diverse signals on both sides of cell membrane. Motile cilia are responsible for cellular swimming and fluid flow. Given the array of diverse functions carried out by centrioles and basal bodies, it is not surprising that mutations affecting centrioles and basal bodies cause diverse human diseases, ranging from tumors to different forms of ciliopathies (*Nigg and Raff, 2009*; *Reiter and Leroux, 2017*)

Centriole biogenesis is tightly controlled. Recent studies have illuminated the mechanisms that regulate centriole assembly. Genetics and cell biological studies in various model organisms show that centriole biogenesis occurs by a cascade of molecular events performed by evolutionarily conserved components, reviewed by *Banterle and Gönczy (2017)*. In vertebrate cells, the sequential recruitment of centriole components during assembly is accompanied by a series of morphological changes at the newly emerging centrioles. First, the nine-fold symmetric structure called the

cartwheel forms at the proximal side of the mature centriole (mother centriole). This is followed by assembly of MT triplets at the tip of the nine radial spokes in the cartwheel. Together, they form the procentriole, a precursor of the daughter centriole. The procentriole continue to elongate and develop as the new daughter centriole prior to mitosis. During mitosis, the mother and daughter centrioles are 'disengaged' but remain loosely linked. In the next cell cycle, the daughter centriole will completely separate from the mother. Meanwhile, it acquires the appendages and the PCM and becomes fully competent as an MTOC. These mark the completion of the centriole duplication cycle (*Kong et al., 2014*). Similar to the centriole duplication in metazoans such as in mammals, the centriole duplication process in the unicellular organism *Chlamydomonas reinhardtii* is tightly controlled. The process has been described in a series of seminal studies (*Cavalier-Smith, 1974*; *Geimer and Melkonian, 2004*; *O'Toole and Dutcher, 2014*). Compared to the vertebrates, despite many morphological and ultrastructural differences in the duplication steps, a number of key components in the *Chlamydomonas* centriole assembly have been found conserved in other organisms (*Dutcher et al., 2002*; *Dutcher and Trabuco, 1998*; *Hiraki et al., 2007*; *Keller et al., 2009*; *Matsuura et al., 2004*; *Nakazawa et al., 2007*). In addition, proteomics and bioinformatics studies in several model organisms have identified a list of major structural components of centrioles (*Andersen et al., 2003*; *Keller et al., 2005*; *Kilburn et al., 2007*; *Li et al., 2004*; *Müller et al., 2010*). Together, these studies concur that the centriole is assembled by a series of evolutionarily conserved protein building blocks. The process is tightly controlled spatially and temporally by a set of regulatory proteins (*Carvalho-Santos et al., 2010*; *Hodges et al., 2010*).

Meanwhile, structural approaches, including super-resolution light microscopy, X-ray crystallography and electron cryo-microscopy, have been applied to put the building blocks into the context of the centriole's 3D structure. Several crystal structures are now available describing components of the centriole, including Plk4, Spd2, Sas6, Cep135, STIL and CPAP. In addition, there have been cryoET studies on assembly of centriole in several organisms (*Greenan et al., 2018*; *Guichard et al., 2010*; *Guichard et al., 2012*; *Li et al., 2012*). In particular, the events of cartwheel assembly has been studied extensively (*Guichard et al., 2012*; *Guichard et al., 2017*; *Hilbert et al., 2016*; *Kitagawa et al., 2011*; *van Breugel et al., 2011*), leading to a molecular mechanism that at least in part establishes the 9-fold symmetry, reviewed in (*Guichard et al., 2018*).

Despite the structural and functional study of many centriole components in the past years, a complete picture of the centriole architecture and its assembly mechanism is lacking. Using cryoET and subtomogram averaging, we describe the triplet structure of the *Chlamydomonas reinhardtii* procentriole. We identify 11 non-tubulin components in the structure that are associated with the triplet tubules in an asymmetric manner. We further present the structure of the A-C linker that laterally bridges neighboring triplets. Finally, using extensive classification and averaging in image processing, we identified two partially assembled triplets at the growing ends to the procentrioles that shed light on the mechanism of triplet and procentriole assembly. Overall, our work presented here builds a framework for understanding the mechanism of centriole biogenesis in molecular details.

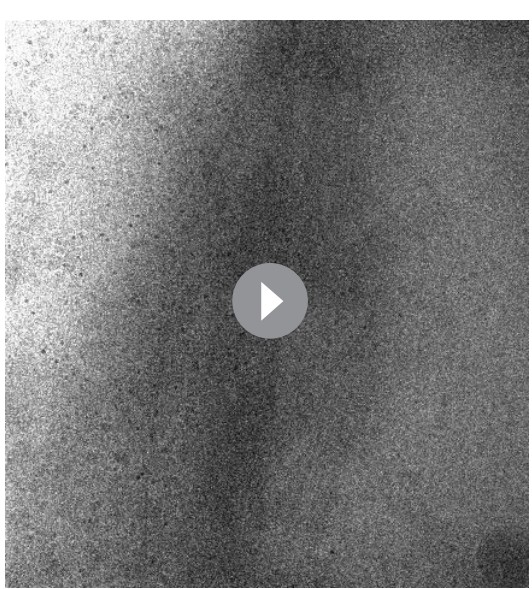

**Video 1.** An Aligned Tomography Tilt Series of NFAp Showing a Pair of Mother Centriole and the Attached Procentrioles.

DOI: https://doi.org/10.7554/eLife.43434.005

## Results

### Overall architecture of the procentriole

To study the structure of both the centriole and procentriole, the nuclear-flagellar-apparatus (NFAp) from the unicellular green algae *Chlamydomonas reinhardtii* were isolated and visualized

by cryoET. The collected tilt series and the reconstructed tomograms show good conservation of the stereotypical structures of NFAp (*Video 1*). These include the proximal and distal striated fibers that connect two mature centrioles, a set of rootlet microtubules (rMT), in the distal end of centriole there is transition zone where it connects to flagellum (*Figure 1A*). At the centriole proximal end, a cartwheel structure, including the central hub and the radial spoke, are visible. In many tomograms, two well-preserved procentrioles are attached to their respective mother centrioles via the rMT (*Figure 1B*, *Video 1*). Here, we focused our study on both the proximal ~100 nm region of mother centrioles and on the attached procentrioles. Using subtomogram averaging, we obtained an averaged structure of MT triplet at 23.0 Å resolution (*Figure 1C*, *Figure 1—figure supplement 1*). In this structure, all MT protofilaments (PF) can be resolved, as well as the 4 nm repeat of tubulin along the PF. This confirms previous observations from the central core region and from centrioles in other organisms that the A-tubule is composed of 13 PFs as an elliptical ring deviated from the canonical MT structure. Both B- and C-tubules are partial rings with 10 PFs. In additional, a number of non-MT structures are readily visible. These include the pinhead, a structure that connects the A-tubule to the cartwheel, and the A-C linker that bridges A-tubule to the C-tubule of its neighboring triplet. To visualize the structure of the entire procentriole, we took reverse steps of subtomogram averaging and put the averaged triplet into the context of entire procentriole tomogram. The result is a model representing a cross section of the procentriole (*Figure 1D*). In the model, without imposing symmetry, the pinhead is about 70 nm away from the central hub. The A-C linker is clearly visible and makes multiple connections to both the neighboring A- and C-tubules, although with blurred details compared to the MTs in the triplet, indicating structural heterogeneity and flexibility in this region.

To assess whether there are structural difference between the proximal region of mother centrioles (10854 subtomograms) and adjacent procentrioles (2083 subtomograms), we averaged these two datasets independently. This results in two triplet averages at 23.0 Å and 30.1 Å resolution respectively (*Figure 1—figure supplement 2*). Even though the MT backbones are nearly identical, there are notable differences between these two averages. A number of microtubule inner proteins (MIPs) that are attached to the centriole B-tubule, in particular MIPs present in the inner junction of A-B and B-C tubule, are absent in the procentriole triplet. Conversely, some luminal densities in the procentriole are missing in the centriole. These differences may reflect difference in kinetics or transient association of proteins at various stages of assembly, or incompletion of procentriole assembly (see below for further detailed analysis on the B- and C-tubules in the procentrioles).

## Non-tubulin components associated with the procentriole triplet

Unlike many biological complexes for which the individual components might have rigid and well-defined structures, large organelle-scale assemblages are often intrinsically flexible and heterogeneous, creating challenges for their structure study. To improve resolution, we applied a 'focused refinement' strategy widely used in single particle cryoEM (*Scheres, 2016*), to align the A-, B- and C-tubules of the triplet separately. These resulted in three averaged tubule structures, with improved resolution at 21.4 Å, 22.3 Å and 22.3 Å, respectively (*Figure 2A–C*, *Figure 1—figure supplement 1*). Based on the improved structures, we built a MT model for each tubule, allowing us to identify a total of 11 MIPs associated with the MT wall (*Figure 2A–C*, *Figure 2—figure supplements 1–11*). Among them, two components MIP1 and MIP2 share similar binding pattern as the MIPs previously seen in the core region (*Li et al., 2012*). MIP1 is a cone-shaped structure projecting from PF A5 into the lumen of the A-tubule and having an 8 nm longitudinal periodicity. Each MIP1 has two legs that recognize the luminal side of α/β tubulin dimer (arrowheads in *Figure 2—figure supplement 1*). By contrast, MIP2 is more filamentous, forming a trellis-like meshwork, laterally spanning PFs A11 to A13. It follows the pitch of the 3-start helical MT lattice running along the A-tubule luminal wall (*Figure 2—figure supplement 2*), presumably to reinforce the 'ribbon' structure that is shared by the A- and B-tubules. Besides MIP1 and MIP2, the overall pattern of the MIPs bound within the procentriole is markedly different from the core region. The procentriole MIP binding patterns are shown in detail in *Figure 2—figure supplements 1–11* and their characteristics are summarized in *Table 1*. Of interest, 7 of the 11 MIPs (MIPs 3 ~ 6, MIPs 9 ~ 11) are localized at or in the vicinity of the inner junctions of the A-B or B-C tubules. Among them, the MIP9 at the inner B-C tubule junction, is of particular interest. Even though it forms a filament exhibiting 4 nm periodicity, the structure deviates from a canonical MT protofilament (*Figure 2—figure supplement 12*). Therefore, we assigned it as a non-tubulin protein MIP9. Together, these inner junctions MIPs crosslink

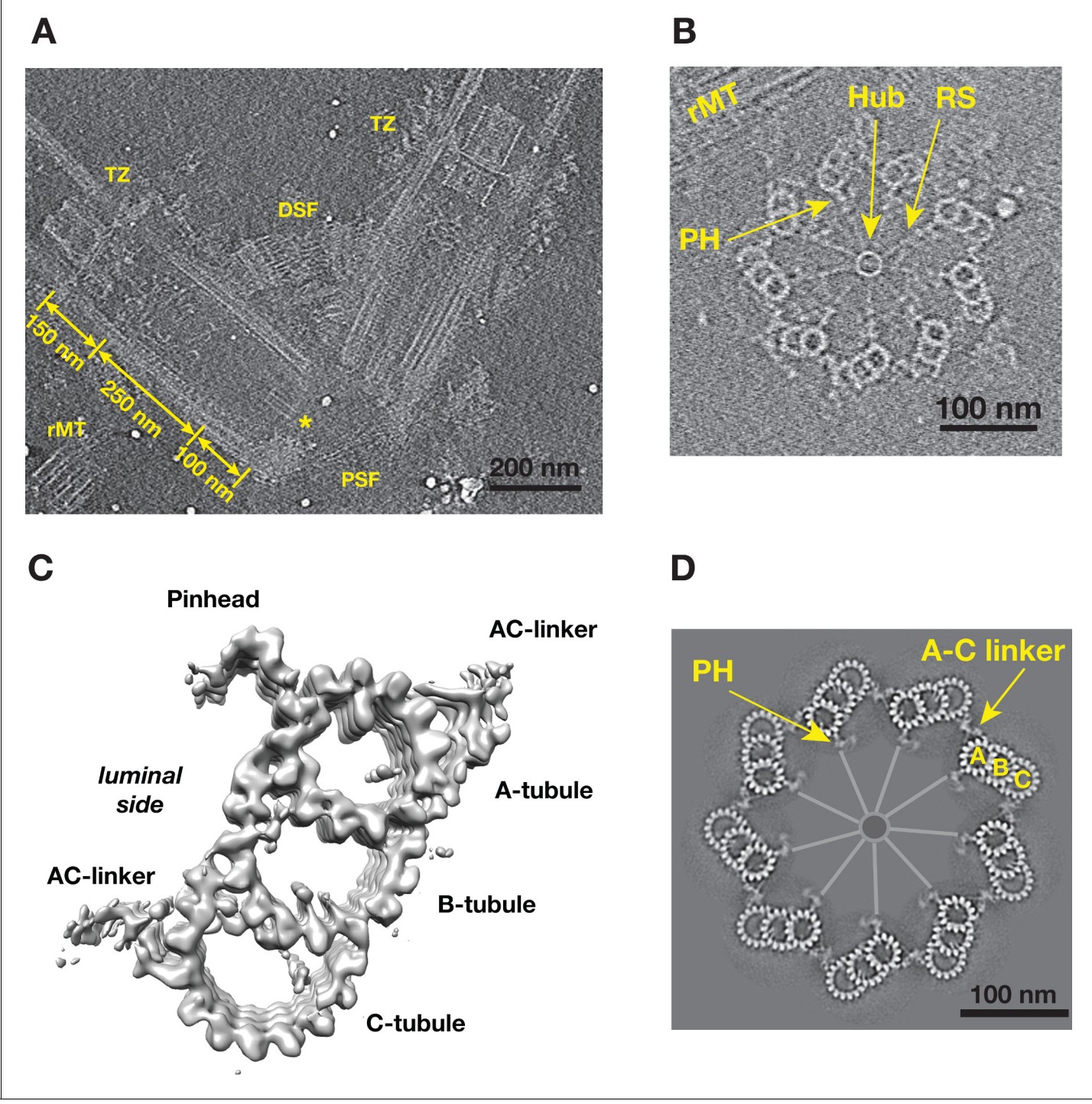

**Figure 1.** Tomographic Reconstruction and Subtomogram Averaging of the MT Triplet in Centriole and Procentriole. (**A**) A slice from the reconstructed tomogram showing a pair of mother centrioles, also known as basal bodies. The central hub as part of the cartwheel is marked by a yellow asterisk. The centriole is partitioned longitudinally into three regions. The procentriole region spans 100 nm at the proximal end of the mother centriole. The central core region spans 250 nm. The distal region spans 150 nm where the triplets become doublets before reaching the transition zone (TZ). PSF: proximal striated fibers; DSF: distal striated fibers; rMT: rootlet MT. (**B**) A slice from the reconstructed tomogram of procentriole attached to the mother centriole via the rootlet MT. The central hub, radial spokes and pinheads are clearly visible, as well as protofilaments in some MT triplets. RS: radial spoke; PH: pinhead; rMT: rootlet MT. (**C**) Subtomogram average of the MT triplet from both the proximal region of mother centrioles and the procentrioles. (**D**) A model of procentriole viewed from its distal end. The model is generated by docking the subtomograms average in (**C**) into the tomogram volume in (**B**). The A- B- and C-tubules and the A-C linker are labeled. The central hub and the radial spokes are depicted schematically. PH: pinhead.

*Figure 1 continued on next page*

*Figure 1 continued*

DOI: https://doi.org/10.7554/eLife.43434.002

The following figure supplements are available for figure 1:

**Figure supplement 1.** Assessing the Resolutions of Subtomogram Average by Fourier Shell Correlation (FSC) Method.

DOI: https://doi.org/10.7554/eLife.43434.003

**Figure supplement 2.** Comparing Subtomogram Averages from a Subset of Data.

DOI: https://doi.org/10.7554/eLife.43434.004

multiple PFs forming intricate networks. They are likely important in strengthening the overall triplet structure.

The cartwheel is a unique procentriolar structure, emerging early in centriole assembly, that is essential in establishing both the 9-fold symmetry and for scaffolding procentriole construction. The overall structure of the cartwheel can be divided into three parts, the central hub, the radial spoke and the pinhead (*Figure 1B*) (*Hirono, 2014*). The pinhead connects the cartwheel to the MT triplets. Using local classification to focus on the pinhead, we identified 4763 subtomograms having relatively intact pinhead structures, resulting in an averaged pinhead structure at 23.1 Å resolution (*Figure 2D*, *Figure 2—figure supplement 13*). It is a hook-shaped structure anchored to PF A3 of the A-tubule. Consistent with a previous study (*Guichard et al., 2013*), the pinhead can be partitioned into three parts, namely pinbody (PinB), pinfoot 1 (PinF1) and pinfoot 2 (PinF2) where both pinfeet attach to the MT wall. Each PinF1/PinF2 unit forms a ring structure bound to α/β tubulin dimer on PF A3, therefore, the entire pinhead repeats every 8 nm. This is in agreement with a recent triplet structure from the proximal region of mammalian centriole (*Greenan et al., 2018*). The PinB turns 90 degree from the pinfeet, forming an inverted L-shaped structure (*Figure 2D*). There is a longitudinal gap between neighboring PinB every 8 nm without connection. The distal tip of PinB is flexible exhibiting less well-defined structure in the average (arrowheads in *Figure 2D* and *Figure 2—figure supplement 14*). Interestingly, among the 13 PFs that form the elliptical-shaped A-tubule, the PFs A2 and A3 display the highest curvature of the MT wall with a large lateral gap between these two PFs where the MIP4 laterally crosslinks. The high local curvature and the gap may create a unique site for anchoring the pinhead to the A-tubule.

## Structure of the A-C linker that connects neighboring triplets

Procentriole is a cylindrical structure composed of 9 MT triplet blades. Each triplet is laterally connected to its neighboring triplets by a structure called the A-C linker. In the averaged triplet structure (*Figure 1C*), the A-C linker is weaker than the MT triplet, indicating either compositional and/or conformational heterogeneity. By extensive subtomogram classification and alignment using the A-tubule as a reference (*Figure 3—figure supplement 1* and detailed in Materials and Methods), we obtained four classes whose averaged structures showing a more complete and detailed structure of the A-C linker most closely associated with the A-tubule (*Figure 3—figure supplements 2,3*). Similarly, in a 'reciprocal classification' by using the C-tubule as the reference point, we identified 6 classes of subtomograms whose averaged structure shows enhanced detail for the portion of the A-C linker associated with the C-tubule (*Figure 3—figure supplements 3–5*). The differences in the classes largely reflect the underlying flexibility of the A-C linker about its midpoint. We combined the above two averages by docking them into a lower resolution class-averages, resulting in a structure of the complete A-C linker for each identified class (*Figure 3—figure supplement 6*).

Overall, the A-C linker forms a crisscross-shaped structure (*Figure 3A*, *Video 2*) that can be divided into five parts: a central trunk region, from which two arms (Arm A, B) and two legs (Leg A, B), extend out to the C- and A-tubules, respectively. Arm A forms a longitudinal helical filament with 8 nm periodicity and binds to PF C8 of the C-tubule (*Figure 3B*). The Arm B inserts into the trough between PFs C9 and C10 of the C-tubule, likely binding to both PFs. Leg A and B fork out from the central trunk towards the A-tubule. The Leg A has a thin rod shape that tilts ~30° towards the proximal end of procentriole and extends about 10 nm to reach and contact PF A6 of A-tubule (*Figure 3C*). Leg B tilts ~30° towards the distal end of procentriole. It spans about 22 nm to connect to both PFs A9 and A10 of the A-tubule. It continues to reach as far as to the outer junction of A-

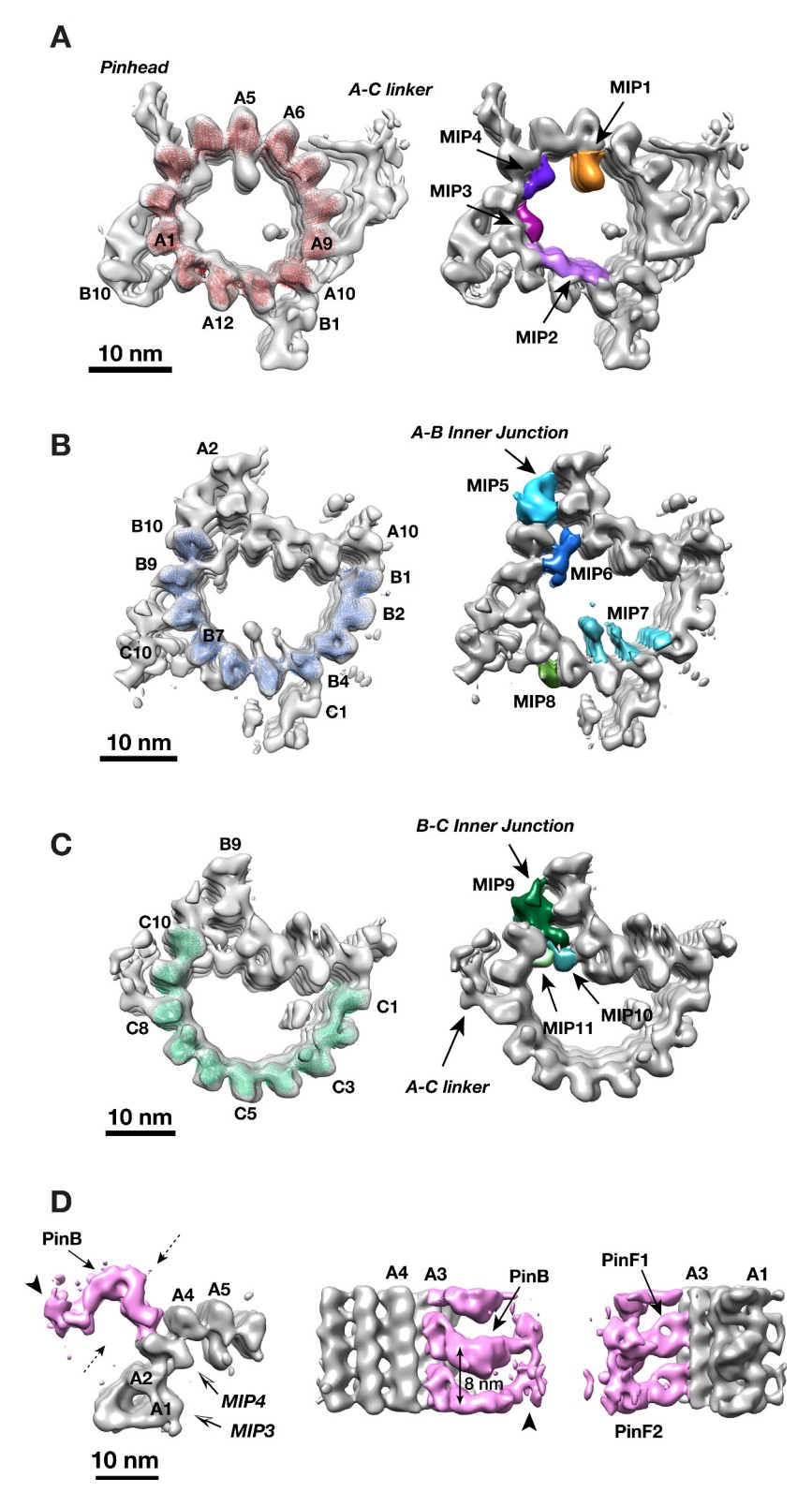

**Figure 2.** Identifying Non-tubulin Components in Procentriole MT Triplet. (**A**) The A-tubule structure. A model of 13-pf MT (red) are docked into the subtomogram average of the A-tubule. 4 MIPs identified on the A-tubule are highlighted in colors. (**B**) The B-tubule structure. A model of 10-pf MT (blue) are docked into the subtomogram average of the B-tubule. 4 MIPs identified on the B-tubule are highlighted in colors. (**C**) The C-tubule structure. A model of 10-pf MT (green) are docked into the subtomogram average of the C-tubule. 3 MIPs found at the B-C inner junction are

*Figure 2 continued on next page*

*Figure 2 continued*

highlighted in colors. (D) Subtomogram averaging of the pinhead structure. The pinhead is highlighted in pink color. Two dashed arrows indicate viewing directions for two images in the side view on the right. The arrows indicate PinB, PinF1 and PinF2, respectively. The arrowheads mark the tip of PinB.

DOI: https://doi.org/10.7554/eLife.43434.006

The following figure supplements are available for figure 2:

**Figure supplement 1.** The non-tubulin Procentriole Components Associated with the MT Triplet.
DOI: https://doi.org/10.7554/eLife.43434.007

**Figure supplement 2.** The non-tubulin Procentriole Components Associated with the MT Triplet.
DOI: https://doi.org/10.7554/eLife.43434.008

**Figure supplement 3.** The non-tubulin Procentriole Components Associated with the MT triplet.
DOI: https://doi.org/10.7554/eLife.43434.009

**Figure supplement 4.** The non-tubulin Procentriole Components Associated with the MT Triplet.
DOI: https://doi.org/10.7554/eLife.43434.010

**Figure supplement 5.** The non-tubulin Procentriole Components Associated with the MT Triplet.
DOI: https://doi.org/10.7554/eLife.43434.011

**Figure supplement 6.** The non-tubulin Procentriole Components Associated with the MT Triplet.
DOI: https://doi.org/10.7554/eLife.43434.012

**Figure supplement 7.** The non-tubulin Procentriole Components Associated with the MT Triplet.
DOI: https://doi.org/10.7554/eLife.43434.013

**Figure supplement 8.** The non-tubulin Procentriole Components Associated with the MT Triplet.
DOI: https://doi.org/10.7554/eLife.43434.014

**Figure supplement 9.** The non-tubulin Procentriole Components Associated with the MT Triplet.
DOI: https://doi.org/10.7554/eLife.43434.015

**Figure supplement 10.** The non-tubulin Procentriole Components Associated with the MT Triplet.
DOI: https://doi.org/10.7554/eLife.43434.016

**Figure supplement 11.** The non-tubulin Procentriole Components Associated with the MT Triplet.
DOI: https://doi.org/10.7554/eLife.43434.017

**Figure supplement 12.** MIP9 forms a non-tubulin filament at the inner junction of B- and C-tubule.
DOI: https://doi.org/10.7554/eLife.43434.018

**Figure supplement 13.** FSC curve for the averaged pinhead structure after classification (resolution: 23.1 Å).
DOI: https://doi.org/10.7554/eLife.43434.020

**Figure supplement 14.** The anisotropic resolution in the pinhead structure.
DOI: https://doi.org/10.7554/eLife.43434.019

and B-tubules (*Figure 3A,B*). Lastly, the central trunk can be further divided into three longitudinal filaments bundled into a spiral: f1, f2, f3 (*Figure 3A,C*).

Strikingly there is a doughnut-shaped density with the Leg B. The averaged diameter of the ring is ~4 nm and they stack longitudinally with 8 nm periodicity (*Figure 3A,B*). Previous studies found the conserved centriole components, POC1, to localize to the centriole proximal ends (*Keller et al., 2009*; *Pearson et al., 2009b*). Knockdown or deletion of POC1 in several organisms resulted in defective centrioles or basal bodies (*Khire et al., 2016*; *Pearson et al., 2009a*; *Venoux et al., 2013*). Human mutations of either paralog, POC1A or POC1B, cause ciliopathy-like pathologies (*Beck et al., 2014*; *Roosing et al., 2014*; *Sarig et al., 2012*; *Shaheen et al., 2012*). In *Tetrahymena*, POC1 is important for maintaining A-C linker integrity. *poc1* null mutants display basal body defects ranging from missing or disintegrated triplets to disconnected neighboring triplets and aberrant A-C linkers (*Meehl et al., 2016*; *Pearson et al., 2009a*). Interestingly, one of predicted conserved signatures of POC1 is multiple WD40 repeats at its N-terminus (*Woodland and Fry, 2008*), for example, in *Chlamydomonas*, POC1 is predicted to have seven tandem WD40 repeats at its N-terminus (*Keller et al., 2009*). A crystal structure of a 7-repeat WD40 β-propeller domain fits remarkably well into the ring density observed on the Leg B (*Figure 3D,E*). Due to relatively modest resolution of our average map, the precise orientation of the WD40 domain could not be defined. Nevertheless, in the docked model, the WD40 β-propeller makes multiple contacts with the rest of Leg B, consistent with the function of WD40 domain as one of the most abundant protein-protein interaction domains that scaffolds multi-protein complexes (*Stirnimann et al., 2010*). Notably, human ciliopathy mutations in POC1A or POC1B map to the WD40 repeats. Thus, combined with these data, our

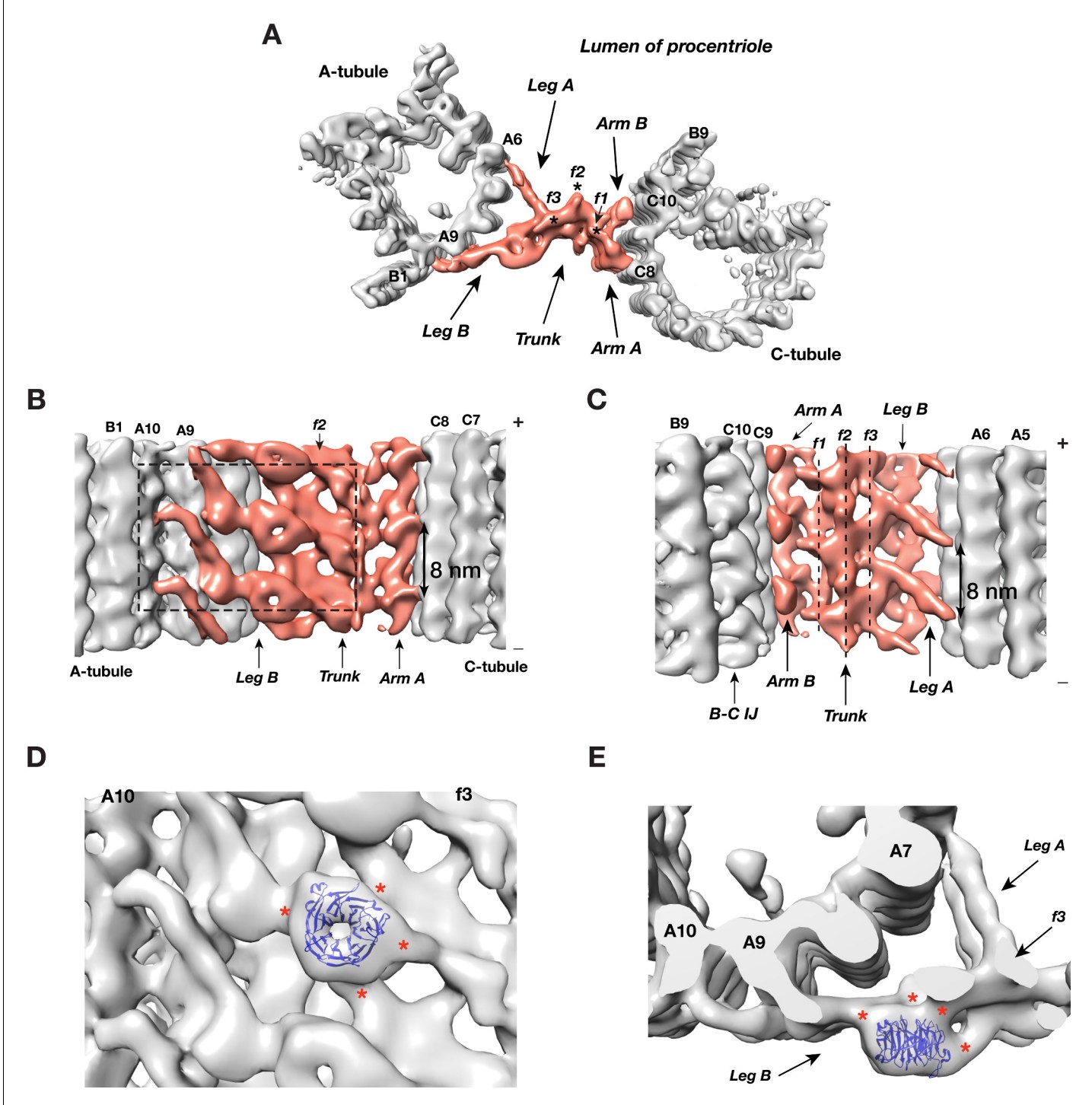

**Figure 3.** The Structure of A-C Linker. (A) A top view of the A-C linker structure with its linked A- and C-tubules. The A-C linker is highlighted in red. Three filamental structures f1, f2 and f3 making up the central trunk running in longitudinal direction are marked with *. (B) The A-C linker viewed from outside of procentriole. The dashed square highlights the area that will be viewed in (D). (C) The A-C linker viewed from the luminal side of procentriole. Three filaments f1, f2 and f3 that make up the central trunk are marked with vertical dashed lines. They are interconnected and form a spiral. (D and E) Close up views of the Leg B in the A-C linker show a possible location of POC1. (D) is in side view from outside of the procentriole. (E) is in top view from the distal end of the procentriole. An atomic model of WD40 β-propeller domain (PDB ID 1S4U) is fitted into the doughnut-shaped density in Leg B. The red * indicate connecting points of this WD40 domain to the rest of Leg B structure and to the f3 in central trunk.
DOI: https://doi.org/10.7554/eLife.43434.021

The following figure supplements are available for figure 3:

*Figure 3 continued on next page*

*Figure 3 continued*

**Figure supplement 1.** Schematic diagram shows the classification process for identifying intact A-C linker associated with the A-tubule.

DOI: https://doi.org/10.7554/eLife.43434.022

**Figure supplement 2.** Subtomogram average of the A-tubule with associated A-C linker.

DOI: https://doi.org/10.7554/eLife.43434.023

**Figure supplement 3.** FSC curves for the averaged A-tubule and the C-tubule with more intact A-C linker after classification.

DOI: https://doi.org/10.7554/eLife.43434.026

**Figure supplement 4.** Schematic diagram showing classification process for identifying intact A-C linker associated with the C-tubule.

DOI: https://doi.org/10.7554/eLife.43434.024

**Figure supplement 5.** Subtomogram average of the C-tubule with associated A-C linker.

DOI: https://doi.org/10.7554/eLife.43434.025

**Figure supplement 6.** The scheme for Reconstruction of Full A-C Linker Structure.

DOI: https://doi.org/10.7554/eLife.43434.027

structure suggests that POC1 is likely an integral component of the procentriole A-C linker that bridges neighboring triplets. Its N-terminal WD40 β-propeller domain provides multiple sites for interacting with other A-C linker components. However, in addition to procentriole, POC1 has also been found at other locations in mother centriole and in flagellum (*Keller et al., 2009*; *Pearson et al., 2009b*). A precise localization of POC1 gene product on the centriole has to wait for future higher resolution structure by cryoET and other studies.

## The twist of A-C linker results in iris diaphragm motion of the procentriole

As discussed above, analysis of the A-C linker heterogeneity led to the identification of 4 and 6 conformational classes in two classification schemes where either the A- or C-tubule was used as a reference point (*Figure 3—figure supplements 1,4*). Overlaying projections from these classes, in both schemes, reveals a large and continuous swinging motion of the remaining portion of the linker relative to the reference point (*Videos 3* and *4*). The swing angle θ (*Figure 4A*, *Table 2*) changes up to 10 or 16 degrees respectively in two schemes. Since the proximal region of the mother centriole has a defined length, calculating the weighted average of the swing angle at each longitudinal position along the triplet (*Figure 4B,C*) shows that two neighboring triplets, by using the A-C linker as a pivot point, twist progressively along the procentriole in a left-handed manner with the thumb pointing towards the MT plus end, as illustrated in *Figure 4A*.

To further identify the origin of this twist motion, we overlaid the complete A-C linker for each class obtained by following steps illustrated in *Figure 3—figure supplement 6*. The two sets of overlaid structures are colored as a 'heat map' with increasing temperature from the proximal to the distal position along the longitudinal axis of the centriole. The overlays clearly show the central trunk region of the A-C linker is at the pivot point of this twist motion (*Figure 4D,E*). Interestingly, the trunk is composed of a bundle of 3 inter-connected filaments, f1, f2 and f3 (*Figure 3A–C*). Together they form a spiral that likely drives, at least in part, the twist motion of neighboring triplets.

Finally, to directly visualize the twist of neighboring triplets in the context of the entire procentriole, we placed the averaged triplet structure back into one of the procentriole tomograms by reversing the steps of subtomogram averaging. The result is a longitudinal segment of the procentriole. Proceeding from the proximal towards the distal end (*Video 5*), the nine triplets rotate concomitantly while the A-C linkers are at the pivot points of the twist. At the distal end, the triplets become more tangential to the circumference of procentriole cylinder. As a consequence, the luminal diameter gradually increases as if opening up an iris diaphragm. Similar diaphragm-like motion of the centriole has been

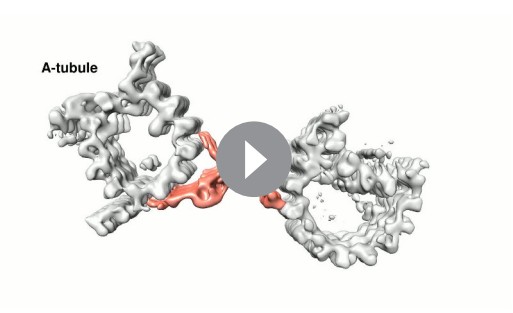

**Video 2.** Surface Rendered Structure of the A-C Linker.

DOI: https://doi.org/10.7554/eLife.43434.028

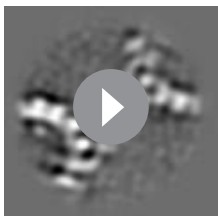

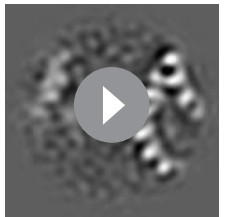

**Video 3.** Swing Motion of the A-C Linker and its Linked C-tubule. The A-tubule on the left is static and is used as a reference point to show the motion.
DOI: https://doi.org/10.7554/eLife.43434.031

**Video 4.** Swing Motion of the A-C Linker and its Linked A-tubule. The C-tubule on the right is static and is used as a reference point to show the motion.
DOI: https://doi.org/10.7554/eLife.43434.032

observed previously in several organisms, including *Chlamydomonas*, *Tetrahymena* and mammalian cells (*Anderson, 1972*; *Li et al., 2012*; *Meehl et al., 2016*; *Paintrand et al., 1992*). Interestingly, a *Tetrahymena POC1* null mutation results in a reduced triplet twist angle (*Meehl et al., 2016*), likely due to defective A-C linker structure. It is likely that this longitudinal twist of triplets is a common structure feature of procentrioles and centrioles. Taken together, our structural analysis of the A-C linker has revealed a characteristic iris diaphragm motion of the procentriole that is accommodated by the twist of a spiral in the central trunk of the A-C linker.

## Intermediate state of the B-tubule during triplet assembly

Many of our tomograms contain procentrioles attached to mother centrioles that are in the process of assembly (*Figure 1B*, *Video 1*). To identify any assembly intermediates and to study the assembly process, we set out to analyze the triplets from these procentrioles. Among 110 tomograms collected, the length of the procentriole triplet varied substantially, with the A-tubule always being the longest tubule, followed by the B- then the C-tubules. Most of the procentriole MTs are straight with a slightly flared end at the MT tip (*Figure 5A*), consistent with the morphology of slow growing MT and elongating triplets (*Guichard et al., 2010*; *Höög et al., 2011*). The B- and C-tubules are attached to the middle of A- or B-tubules at different heights, respectively, suggesting that the B- and C-tubules initiate assembly by using the A- and B-tubule wall as a template rather than a minus-end template such as the γ-tubulin ring complex. Once initiated, the B- and C-tubules then extend longitudinally towards both the MT minus and the plus ends. Histograms of the measured tubule length in 110 procentrioles show wide distributions for all three tubules (*Figure 5B*), demonstrated by the large standard deviations. This length variation also implies substantial structural heterogeneity in the procentriole triplets where the B- and C-tubules are likely partially assembled. We set out to identify any assembly intermediates using classification.

We first focused on classification of the B-tubules. Based on 2083 subtomogram volumes from 110 procentrioles, we obtained eight classes (*Figure 5—figure supplement 1*). Surprisingly, one of the classes, composed of 157 subtomograms, shows partially assembled B-tubules. The average of this class shows three PFs B1, B2 and B3 at outer A-B junction having more prominent density than the other PFs in the B-tubule. We mapped these 157 subtomograms to find their location in the procentriole. Based on their longitudinal positions, we divided them into five classes. These are 1) at the proximal end of the triplet (28.7%), 2) at the distal end (25.5%), 3) a nascent B-tubule where the entire B-tubule is partially assembled (33.8%), 4) extremely short and noisy B-tubule less than 10 nm long (7.0%), 5) partial B-tubule structure found in the midst of complete tubule (5.1%), likely these are defective structures or errors in classification. The first three classes comprise 88% of the 157 subtomograms, and the averages are shown in *Figure 5C*. Interestingly, all three averages show an incomplete B-tubules with strong densities for PFs B1 ~B3. As a check, we directly visualized the identified partial doublets in their procentriole tomograms (*Figure 5D*), confirming that they are indeed partially assembled doublets at the longitudinal extremity of the triplet. In summary, based on the classification and the mapping analysis, we identified an intermediate in the B-tubule assembly at both the polar ends of triplet. It shows the B-tubule initiates lateral expansion from PF B1 at the outer A-B junction towards the luminal side.

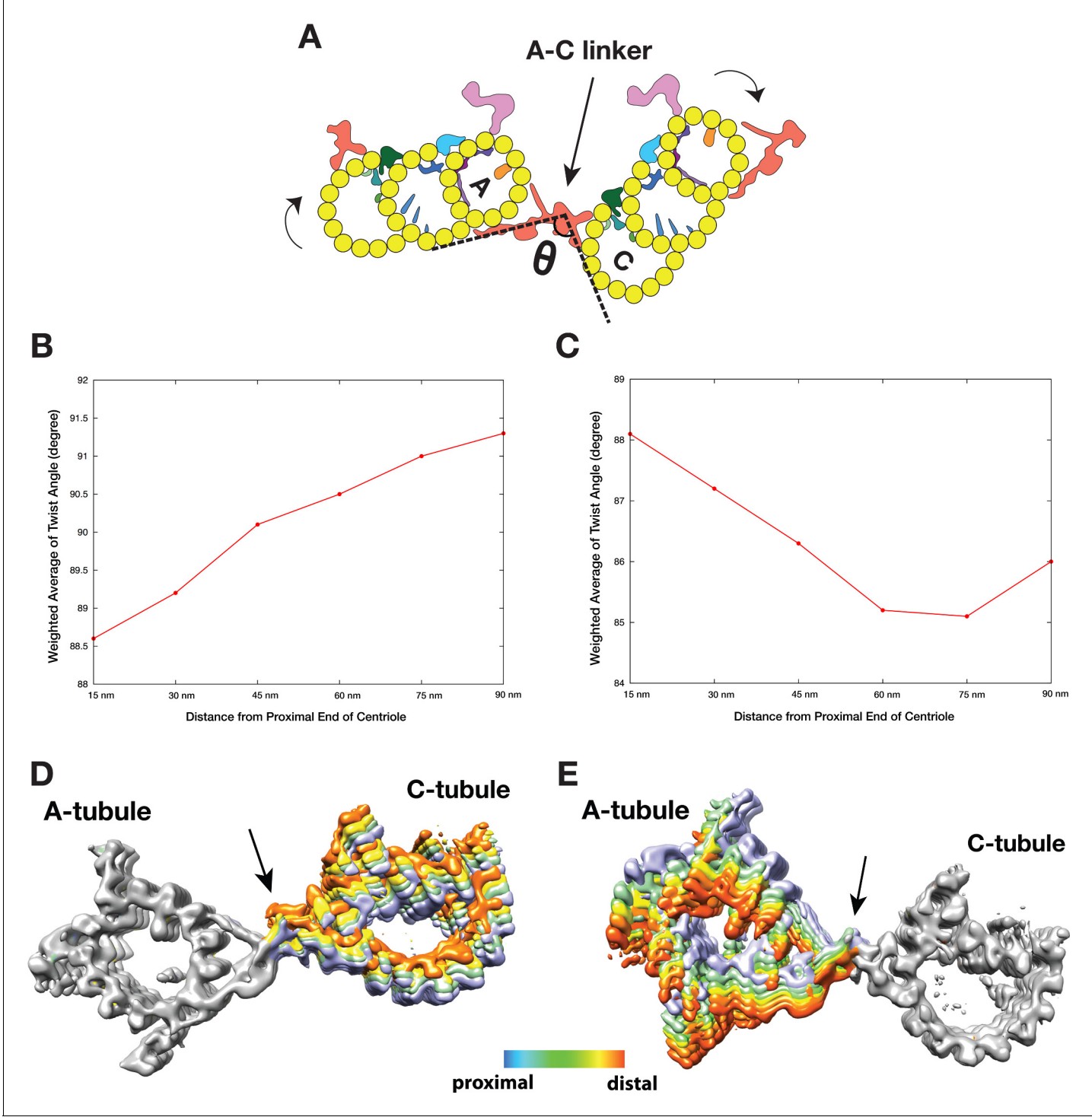

**Figure 4.** The Twist of A-C Linker Drives Procentriole Triplets in Iris Diaphragm Motion. (**A**) Schematic diagram of two triplets connected by the A-C linker. θ is defined as the angle between the Arm A and the Leg B. The results of mapping and angle measurement as shown in (B-C, **Table 2**) are consistent with a twisting motion of two triplets as indicated by two curved arrows when moving longitudinally from proximal to distal direction. The pivot point of this twist is at the A-C linker as indicated by an arrow. (**B and C**) Weighted averages of twist angle along the triplet wall in two classification schemes shown in **Figure 3—figure supplements 1,4**. Six points along 90 nm longitudinal length of centriole are sampled starting from the proximal end. The weighted average of twist angle T at point $i$ is defined as: $T_i = \sum \theta_j * N_{ij} / \sum N_{ij}$ . $N_{ij}$: number of subtomogram belong to class $j$ at point $i$. $\theta_j$: twist angle as defined in (**A**) for class $j$. (**D and E**) Overlay of two sets A-C linker structure based on two classification schemes (**Figure 3—figure supplements 1,4**). Based on its longitudinal position, each structure is colored following a 'heat map' scheme. Arrows point to the pivot point, the central trunk.

*Figure 4 continued on next page*

*Figure 4 continued*

DOI: https://doi.org/10.7554/eLife.43434.029

## Intermediate state of the C-tubule during triplet assembly

We applied a similar analysis to the C-tubule and after two rounds of classification, we identified 208 subtomograms from 110 procentrioles that have partially complete C-tubules (*Figure 6—figure supplement 1*). In contrast to the incomplete doublet, the average of these 208 subtomograms shows PFs at both the inner and the outer B-C junctions with a gap in between. We further mapped these incomplete triplets in their corresponding tomograms. Similar to the B-tubule result, based on the location these triplets can be divided into five classes. Three major classes, including the proximal end, the distal end and the nascent C-tubule where the entire length of the triplet has partially assembled C-tubule, comprise 83% of the 208 subtomograms. Averages from these three major classes shows a consensus structure of incomplete triplet (*Figure 6A*), where the PFs C8 ~C10 along with the MIP9 at the inner B-C junction are clearly visible while the PFs C1 ~C3 are visible at the outer junction. Interestingly, in all three averages, even though the C-tubule is incomplete, the A-C linker is visible albeit with weaker density likely due to its flexible nature or incomplete assembly. This indicates that the linkage between neighboring triplets has been established at this early stage of the triplet assembly. We further confirmed these averages by directly visualizing corresponding structure in the procentriole tomograms (*Figure 6B*). Taken together, our analysis has identified C-tubule assembly intermediates at either end of the triplet longitudinal extremity, suggesting that the C-tubule assembles laterally from both the inner and the outer junction in a bi-directional fashion. This allows prompt establishing of the A-C linker during the triplet assembly, reinforcing the procentriole as a barrel-shaped structure.

## Discussion

Here, by using cryoET and image analysis, we present procentriole structures from *Chlamydomonas reinhardtii*. These new structures and findings have implications concerning the mechanism of centriole biogenesis and assembly.

## Comparing the chlamydomonas procentriolar triplet to other triplet structures

As important cell organelles, there have been extensive electron microscopic studies on both the procentriole and the centriole. Most were focused on the morphology of these organelles in the context of their life cycles. In recent years, cryoET has been applied to study molecular architecture of centriole and basal body in several organisms, including *Chlamydomonas*, *Drosophila*, *Trichonympha* and mammalian cells (*Greenan et al., 2018*; *Guichard et al., 2013*; *Li et al., 2012*). These have provided insights into mechanism of triplet/doublet assembly and centriole biogenesis.

In *Chlamydomonas*, there are marked differences between the procentriole triplet structure reported here and that in the core region (*Li et al., 2012*). First, a Y-shaped structure observed in the luminal side of core region is absent in procentriole, instead, the space is taken in part by the pinhead structure. Second, the A-C linker structures in these two regions are different. Whereas the procentriole has a crisscross shaped A-C linker structure that links PFs A6, A9/A10 and C8, C9/C10 from its neighboring triplets, in the core region, the A-C linker bridges PF A6 to various positions on neighboring C-tubule depending on the longitudinal position (*Li et al., 2012*). Third, the PF C1 in the procentriole is composed of tubulin having a 4 nm periodicity, while the C1 in distal half of the core region exhibits an 8 nm repeat, indicating a non-tubulin protein. Lastly, even though a number of MIPs are shared in both regions, for example a cone-shaped structure MIP1 at PF A5 and a number of MIPs spanning the luminal side of PFs A1-A3, the overall patterns of MIPs decorated on the triplet are different in these two regions. This is particularly noticeable in the inner junctions of A-B and B-C tubules. All these differences suggest that two sets of proteins are enlisted sequentially during the centriole assembly, controlled by as yet unknown mode of spatial and temporal regulation. For example, both pinhead and the Y-shaped structure are near the inner junction of A-B tubules. As part of the cartwheel structure, the pinhead function as a scaffold for initial establishment of nine

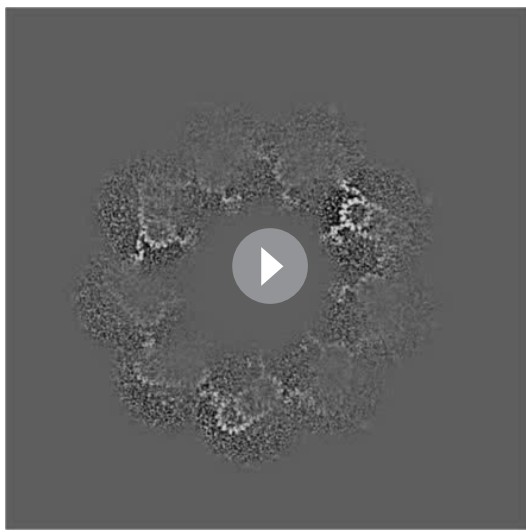

**Video 5.** An Iris-Diaphragm Motion of the Procentriole Triplets. A longitudinal segment of the procentriole is generated by putting the averaged triplet structure back into one of the procentriole tomograms. The movie is displayed as moving through the longitudinal direction from the proximal to the distal end then backwards. The triplets twist progressively along the procentriole in a left-handed chirality with the thumb pointing towards the distal end. Two yellow arrows mark the two A-C linkers. They appear at the distal end of the longitudinal segment.
DOI: https://doi.org/10.7554/eLife.43434.033

triplets in the procentriole. Once the procentriole is fully assembled, the Y-shaped structure will follow to further strengthen the central core region. Structural difference between these two regions has also been observed recently in mammalian centriole (*Greenan et al., 2018*), but with details that are distinct from those with *Chlamydomonas*. However, this does suggest that a multiple-step hierarchical spatial and temporal control is likely a common theme of centriole assembly in many eukaryotes.

The procentriole triplet structure presented in this work also exhibits several notable differences from the triplets from other organisms. For example, in *Trichonympha*, the pinhead exhibits 17 nm periodicity, while in *Chlamydomonas*, it is 8 nm. Interestingly, in both organisms the A-C linker has 8.5 nm or 8 nm periodicity, respectively. This structural difference might reflect the difference in mechanism of length control as the procentriole in *Trichonympha* has an exceptionally long cartwheel.

## The roles of A-C linker in the procentriole assembly

In addition to serving as a structural linkage between neighboring triplets, our structure has revealed several important roles for the A-C linker during procentriole assembly.

In a *Sas6* null mutation in *Chlamydomonas*, a circular centriole can still form without the cartwheel, albeit at low efficiency (*Nakazawa et al., 2007*). Among these cartwheel-less centrioles, the majority remains nine-fold symmetric, but other symmetries have been observed as well, suggesting that, at least in some organism, the cartwheel does not fully account for setting up the nine-fold symmetry in the centriole. Interestingly, in vitro assembly of CrSas6 results in cartwheel-like structures with the majority having eight and nine-fold symmetry (*Guichard et al., 2017*; *Hilbert et al., 2016*). Likewise, mammalian cells can form centrioles de novo without the central hub formed by Sas6, although the assembly is less efficient and more error-prone than the wild type (*Wang et al., 2015*). These observations have led to speculation that other factor(s) might be needed for efficient and robust centriole assembly (*Gönczy, 2012*; *Hirono, 2014*). Our finding that the A-C tubule linkage is established at an early stage of procentriole assembly while the triplets are still elongating provides direct evidence supporting a model that the A-C linker helps establish the 9-fold symmetry. In our model, A-C linker can accomplish this simply by restricting the angle between neighboring MT triplets to be approximately 140 degree as needed for forming a nonagon. Thus in wild type cells, the combination of both Sas6 oligomerization and a stabilizing linkage between neighboring triplets that has a defined angle will provide a robust mechanism for establishing the nine-fold symmetry. In addition to restricting the angle between neighboring triplets, the A-C linker also plays a structural role that connects nascent incomplete triplets therefore bolstering the nascent procentriole structure during its assembly.

Instead of being a static structure, the A-C linker displays substantial twist motion along the triplet wall. This can accommodate an apparent left-handed iris diaphragm movement that progresses along the longitudinal direction. The twist observed both in the procentriole and in the core region results in change of angle for each triplet relative to the cylindrical circumference, from the most acute angle at the very proximal side towards more tangential in the distal region. As a result, the inner diameter of the centriole gradually increases. This change may facilitate multiple processes: First, the acute angle of the triplet at the proximal end places the A-tubule including the pinhead

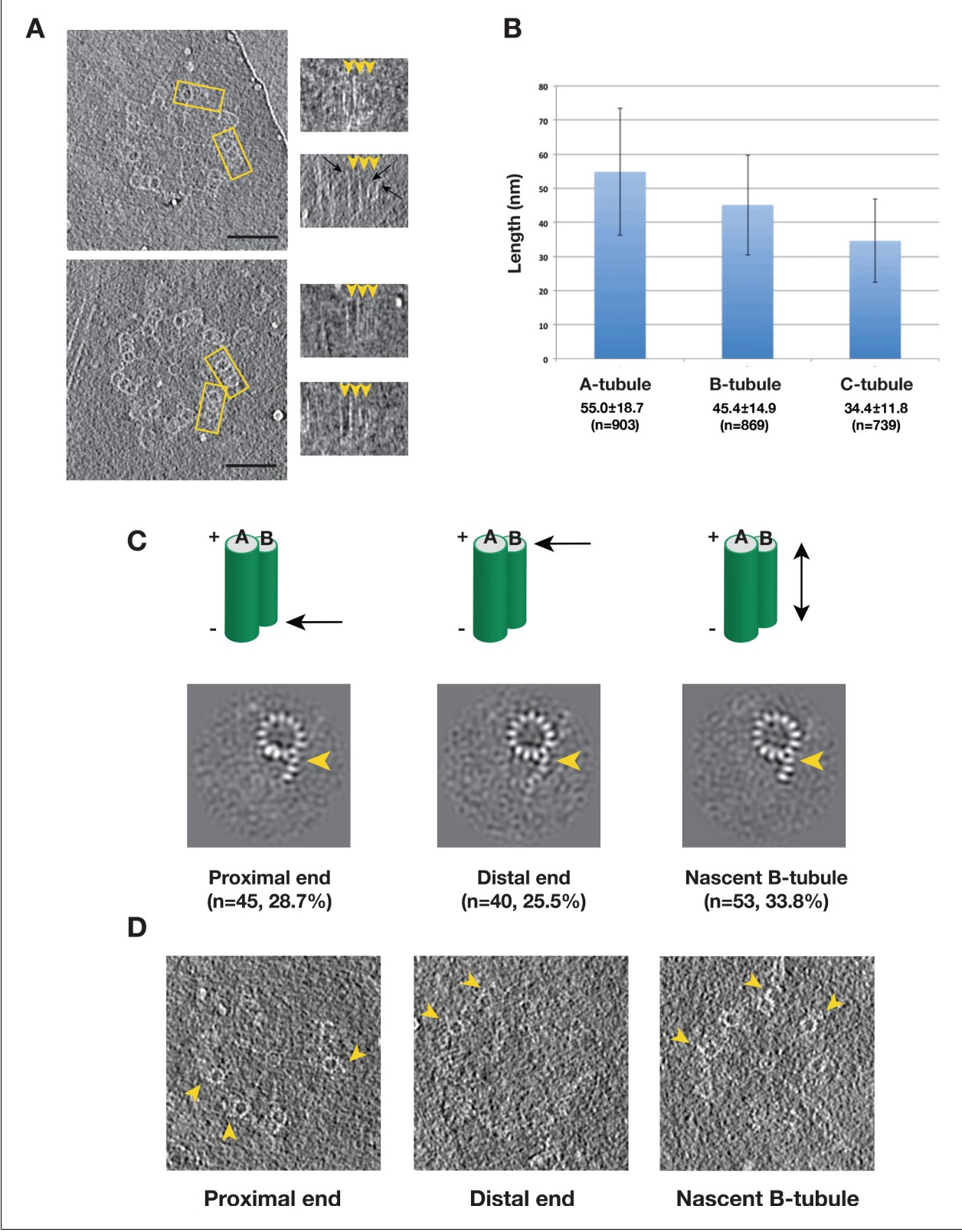

**Figure 5.** Variation of the Tubule Length in Procentriole Triplets and Detecting the B-tubule Intermediate. (**A**) Two examples of procentriole in tomogram. In each, two triplets marked by yellow rectangle box are selected to depict tubule length variation as shown on the right in the longitudinal view. The distal end of each tubule is indicated by an arrowhead. Black arrows indicate examples of the slightly flared morphology at the growing end of the tubule. Scale bar: 100 nm. (**B**) Histogram showing length distribution for A-, B- and C-tubules. For each tubule, the average length, its standard

*Figure 5 continued on next page*

*Figure 5 continued*

deviation and the number of measurement are indicated. (**C**) The averages of incomplete B-tubule from the proximal, distal and nascent doublet, respectively. Each image is a z-projection of doublet of 9.6 nm long. The numbers of subtomogram in each class and their percentages are indicated. The arrowhead indicates PF B1. (**D**) Examples of incomplete B-tubule in procentriole tomogram from three classes. The incomplete B-tubules are marked by yellow arrowheads.

DOI: https://doi.org/10.7554/eLife.43434.034

The following figure supplement is available for figure 5:

**Figure supplement 1.** Classification and Identification of Partially Assembled B-tubule.

DOI: https://doi.org/10.7554/eLife.43434.035

close to the cartwheel, likely reflecting its attachment as the earliest stages of biogenesis. Second, the gradual twisting will increase the exposed surface area, facilitating recruitment of other centriole components, such as the subdistal and the distal appendages. Third, the tangential angle of the triplet at the distal end of basal body must match the angle of the axoneme doublets, where the basal body is ready to serve as a template for the flagellum assembly. Lastly, this iris diaphragm motion of the MT triplet might also have impact on controlling the procentriole length by twisting the pinheads away from the hub while the luminal diameter becomes wider. This suggests that the radial spokes will be under strain of pulling force that continually increases along the procentriole. This pinhead movement can be directly visualized in *Video 5*. As part of the radial spoke, the C-terminus of Sas6 and Cep135/Bld10 are predicted to have highly coiled-coil structures with intrinsically elastic properties. It is appealing to envision that this strain and elasticity will reach to a balance point once the MT triplets elongate to certain length, suggesting an intrinsic length control mechanism due to the iris diaphragm motion of the triplet.

## A model for the procentriole triplet assembly

Previous EM studies on mammalian centriole and basal body showed the triplet assembly was a non-uniform process where the A-tubule assembled first, followed by sequential B- and C-tubule assembly. During the tubule elongation, their lengths varied substantially (*Anderson and Brenner, 1971*; *Guichard et al., 2010*). Furthermore, by analyzing the nascent procentrioles, Anderson and Brenner observed the B- and C-tubule first emerged as sheets laterally attached to the A- and B-tubule respectively. Consistent with these findings, by analyzing a population of procentrioles, we found the average length of three tubules in the triplet varies substantially. In addition, we have identified two triplet intermediates, showing that the B-tubule initiates assembly uni-directionally from the outer A-B junction. By contrast, the C-tubule assembles laterally in a bi-directional fashion, from both the inner and the outer B-C junctions. These lateral expansions are concomitant with the longitudinal elongation of the triplets at their extremities. Based on the analysis, we propose a possible model for the triplet assembly during the procentriole biogenesis (*Figure 7*). In our model, the A-tubule first emerges and anchors to the pinhead of cartwheel. It continues to grow. Then the B-tubule initially assembles at the outer A-B junction by making an unusual tubulin-tubulin lateral interface and branching out at the PF A10. This branch laterally expands from the outer junction to reach the luminal inner junction where the PF B10 and other components including the MIPs 5 ~ 6 seal and complete the B-tubule. Unlike the B-tubule, the C-tubule initiates assembly on both junctions. PFs C1 and C10 branch out from the B-tubule at the outer and the inner junctions, respectively. These two branches expand laterally in a bi-directional fashion. At this point, while the C-tubule is still incomplete, the newly assembled PFs C8 ~C10 are readily available to provide a site for the A-C linker that connects the C-tubule to the A-tubule of its (N-1) neighbor triplet. Equivalently, its A-tubule will connect to the C-tubule of its (N + 1) neighbor triplet. As a result, all nine triplets form an inter-connected barrel even though some triplets are still incomplete. In the last step, the gap is filled by the PFs in the C-tubule. This completes the assembly of all three tubules at this longitudinal point, while the lateral expansion and the longitudinal elongation will continue in both extremities of the triplet.

Unlike the rapid kinetics of the MT self-assembly, the dynamics of growing MT triplets in both procentriole and centriole are slow but stable compared to the MT (*Kochanski and Borisy, 1990*). Although we cannot exclude the possibility that there is a fraction of partial B- or C-tubules that is due to their lability during the purification process, based on the observed tubule length variation,

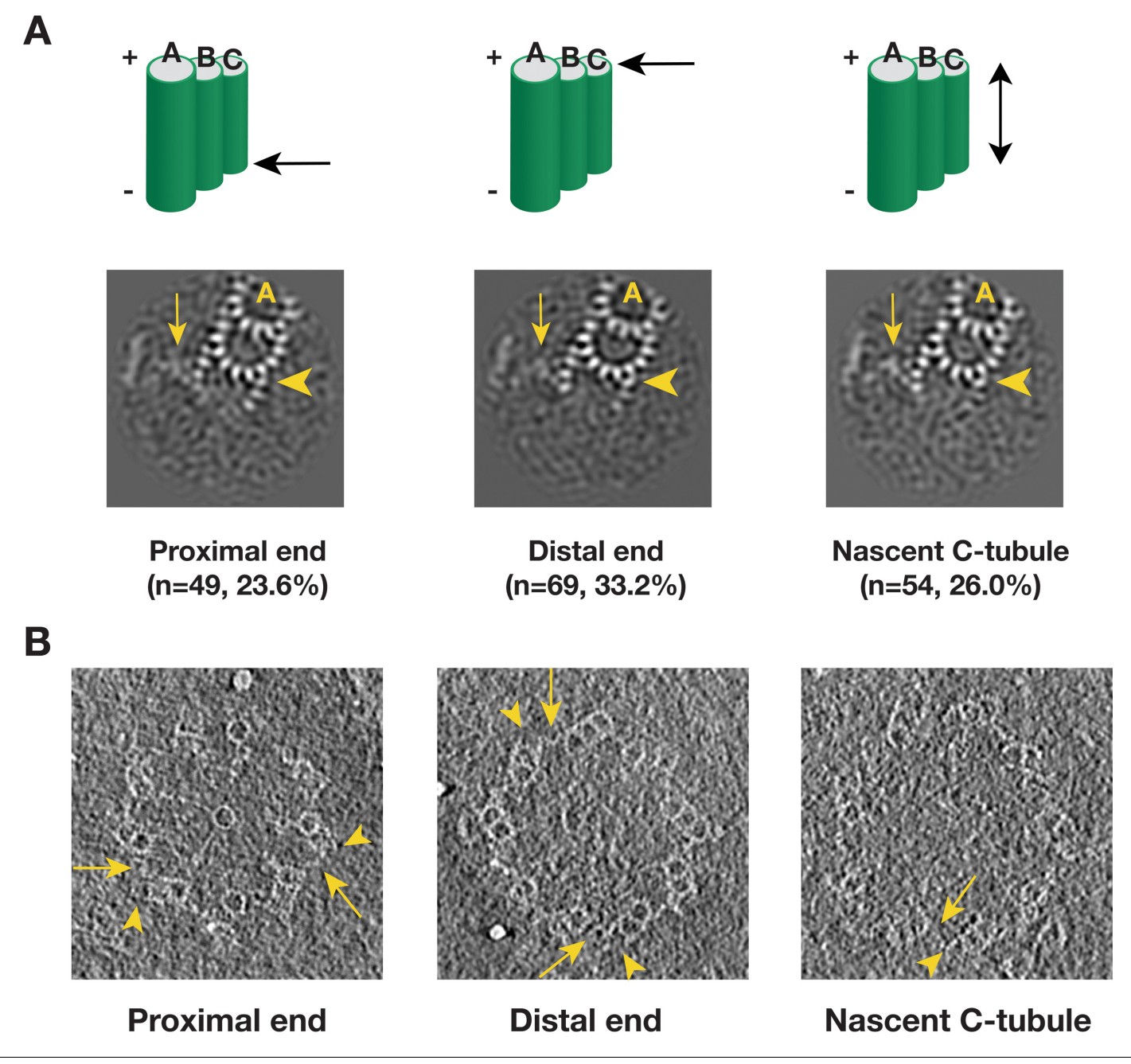

**Figure 6.** Detecting the C-tubule Intermediate. (**A**) The averages of subset of subtomogram with incomplete triplet at proximal, distal end and from nascent triplet. The images are z-projection of the average where the triplet length is 9.6 nm. The numbers of subtomogram in each class and their percentages are indicated. The arrowhead points to PF C1. The arrow points to the A-C linker. (**B**) Examples of incomplete C-tubule in procentriole tomogram in three classes. The incomplete C-tubules are marked by yellow arrowheads. The A-C linkers are marked by yellow arrows.

DOI: https://doi.org/10.7554/eLife.43434.036

The following figure supplement is available for figure 6:

**Figure supplement 1.** Classification and Identification of Partially Assembled C-tubule.

DOI: https://doi.org/10.7554/eLife.43434.037

the morphology of the MT ends, the stability of centriole MT compared to the canonical MT and the consistent pattern of the partial B- and C-tubule observed at the growing ends (*Figures 5C–D* and *6A–B*), the identified intermediate structures likely represent two snapshots of an underpinning

mechanism for the MT triplet assembly, which is tightly controlled by a set of factors. These include structural components such as MIPs identified in this work or other microtubule associated proteins (MAPs) (*Sharma et al., 2016*; *Zheng et al., 2016*). In addition, other regulatory components such as, acetyltransferase or glutamylase will alter the polymerization properties and stability of the MT triplet as well (*Yu et al., 2015*). Interestingly, many MIPs identified in this work and in mammalian centrioles (*Greenan et al., 2018*) are localized at the junction of A-B or B-C tubules in the triplet. For example, MIPs 5 ~ 6 are found at the inner A-B junction, while MIPs 9 ~ 11 are at the inner B-C tubules junction. This asymmetric localization of the MIPs will likely modulate the assembly kinetics of the MT PFs. Furthermore, since the association of MIPs to triplet might be concomitant with the MT elongation, the MIPs' own assembly kinetics on MT will in turn have directly impact on the assembly of the triplet. In the future, with enhanced dataset size and improved algorithm, it is likely that more assembly intermediates will be identified in *Chlamydomonas* and in other organisms. This will provide insight into the centriole assembly mechanism with improved spatial and temporal resolution. Our study presented here has paved the way in this direction.

Finally, like many large and functional biological assemblages, the procentriole structure presented in this work is an example demonstrating the extensive diversity and heterogeneity of a biological system in situ, including composition heterogeneity such as the incomplete B- and C-tubules in triplet and conformation heterogeneity such as the continuous twisting of the A-C linker. CryoET is well poised as a unique and unparalleled tool to reveal this phenomenon and to provide high-resolution insights for many biological mechanisms waiting to be explored.

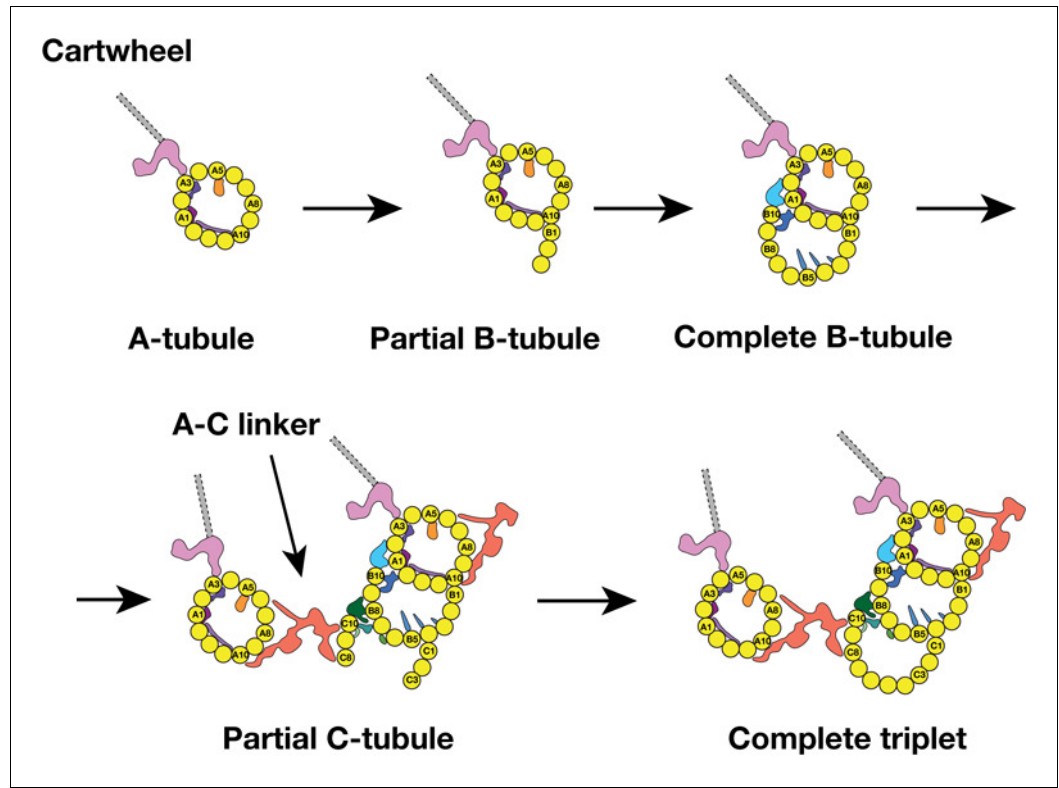

**Figure 7.** A Model for the MT Triplet Assembly in Procentriole. In the model, the triplet assembly can be divided into five sequential steps: 1) A-tubule emerges and anchors to the pinhead in the cartwheel. It continues to elongate longitudinally. 2) B-tubule branches out at PF A10 in outer A-B junction. 3) B-tubule expands laterally from outer A-B junction toward the luminal inner A-B junction where it completes the B-tubule. 4) C-tubule branches out at both inner and outer B-C junctions. Meanwhile, the A-C linker is established. 5) The bi-directional lateral expansion of C-tubule completes the triplet assembly at this longitudinal position.
DOI: https://doi.org/10.7554/eLife.43434.038

## Materials and methods

### Sample preparation

Nucleo-flagellar apparatus (NFAp) from *Chlamydomonas reinhardtii* was purified based on previous published method (*Wright et al., 1985*). Briefly, cells of a cell-wall defect strain (cc849, cw10 mt-, *Chlamydomonas* Resource Center) were grown in TAP medium. Cultures were bubbled continuously with filtered air and illuminated with continuous white light. After reaching log phase, the cells were harvested by centrifugation at 250 g for 5 min. The cells were resuspended in MT Buffer (30 mM Tris-acetate, 5 mM MgSO$_4$, 5 mM EGTA, 25 mM KCl, 1 mM DTT, pH = 7.3). The suspension was overlaid on a 30% Percoll (GE Health) in MT Buffer. The sample was centrifuged for 5 min at 300 g. The supernatant was discarded. The pellet was resuspended in MT buffer, placed on ice, equal volume of Lysis Buffer (MT Buffer, 1% Nonidet P-40 (Roche), EDTA-free protease inhibitor cocktails (Roche) was added, mixed vigorously. The cells were lysed rapidly and completely in 10 min.

The lysate was overlaid on top of discontinuous gradient of 50%, 40% and 30% Percoll in MT Buffer, spun at 10,000 g. Fractions in 30% Percoll was recovered and diluted 10-fold with MT buffer. The sample was pelleted at 800 g followed by resuspension in a small volume of MT Buffer.

### Cryo-ET data collection and reconstruction

The purified NFAp was mixed with BSA-coated 10 nm colloid gold (Ted Pella). 4 µl sample was applied onto Quantifoil grid Rh/Cu 200 R2/2 (Quantifoil, Inc), blotted for 2 s in 100% humidity and flash-frozen in liquid ethane using a Vitrobot (FEI, Inc).

Tomography tilt series were collected on a field emission gun (FEG) microscope (Polara, FEI, Inc) operated at 300kV. The microscope was equipped with a post-column energy filter Bio-Quantum GIF (Gatan, Inc). The slit width was set at 25 eV. UCSF Tomography software (*Zheng et al., 2007*) was used for collecting single-axis tilt series. For screening condition purpose, initial tomographic tilt series were collected on a CCD camera (UltraCam, Gatan Inc). But all tilt series used for subtomogram averaging and for classification were collected on a K2 direct electron detector in the counted mode (Gatan, Inc), in a nominal magnification of 50,000, the effective pixel size is 4.82 Å. The specimen was tilted from −60° to +60° in 1° increment. The exposure time at each tilt was 1.0 s with 0.2 s per frame. This resulted in five frames at each tilt. The dose rate was set at eight electron/pixel/second. The total accumulative dose for each tilt series was about 60 electron/Å$^2$. The defocus was set in the range of 3 ~ 6 µm.

For tomogram reconstruction, a movie of 5 frames at each tilt was corrected of motion using MotionCorr (*Li et al., 2013*). Tomographic tilt series were aligned in IMOD (*Kremer et al., 1996*) by using 10 nm colloid gold beads as fiducials. The contrast transfer function for each tilt series was determined and corrected by TOMOCTF (*Fernández et al., 2006*). 3D tomograms were calculated by TOMO3D (*Agulleiro and Fernandez, 2015*).

### Subtomogram averaging and classification

From 193 reconstructed tomograms containing NFAps, 201 centrioles and 110 procentrioles were identified and selected for further processing. They were first annotated to find the proximal end of MT triplet. Based on these, the longitudinal spans of the MT triplet were defined. The subtomograms that contained segment of MT triplet were boxed out. Initially the triplet segment length was set to 38 nm with pixel size in 2xbinned format. This was followed by *ab initio* alignment without a reference. The alignment of subtomograms was carried out by the programs MLTOMO implemented in the software package Xmipp (*Scheres et al., 2009*). After generating the initial reference, the subtomograms were re-extracted from the tomogram volumes in unbinned format. The MT triplet length in each subtomogram was gradually decreased. The final triplet length in the averaged structures for triplet, A-, B- and C-tubules was 29 nm, containing more than three repeats as the largest periodicity detected in the procentriole triplet is 8 nm. Since MT triplet is a continuous filament, the neighboring subtomograms should have similar geometric parameters. We have developed a program called RANSAC to impose constraint on the neighboring subtomograms from the same triplet. It was used to detect any alignment outliers and to impose constraint to correct misaligned subtomograms by regression. The program is available upon request. The program MLTOMO was also used in the classification of various structures such as the pinhead, the A-C linker,

**Table 3.** Summary of Statistics of Subtomogram Averages

| Structure | Number of subtomogram | Resolution (Å) | Description | EMDB ID # |
|---|---|---|---|---|
| 1 | 12937 | 23.0 | Triplet structure | 9167 |
| 2 | 12517 | 21.4 | A-tubule | 9168 |
| 3 | 12179 | 22.3 | B-tubule | 9169 |
| 4 | 12075 | 22.3 | C-tubule | 9170 |
| 5 | 4763 | 23.1 | Pinhead structure with its associated partial A-tubule | 9171 |
| 6 | 3992 | 23.1 | A-tubule with more complete A-C linker after classification | 9172 |
| 7 | 3245 | 23.1 | C-tubule with more complete A-C linker after classification | 9173 |
| 8* | Composite map derived from EMD-9172 and EMD-9173 | | Full A-C linker structure with A- and C-tubule it links to | 9174 |

All resolutions are assessed by using the 'gold standard' scheme with FSC 0.143 criterion. * This is a composite map by merging EMD-9172 and EMD-9173 onto the map of one of classes, class#2 (See **Figure 3—figure supplement 6**)

DOI: https://doi.org/10.7554/eLife.43434.039

the A-tubule, B-tubule and C-tubule to detect heterogeneity. The processes are illustrated schematically in the supplementary figures.

For each tubule structure from 110 procentrioles, we first carried out focused classification on 2083 subtomograms on the A-, B- and C-tubule. This allowed us to identify 420, 758 and 862 defective or incomplete structure in each tubule respectively. These subtomograms were excluded from further focused alignment. The results are more homogeneous datasets for each tubule that combine procentriole and the proximal region of the mother centriole. The improvements are evident in the increased resolutions shown in *Figure 1—figure supplement 1* and in *Table 3* for all three tubules. These averaged tubule structures were used for further segmentation and estimation of the molecular weight for each MIP associated with the tubule.

For all the structures reported, resolution assessments were carried out in a 'gold-standard' scheme (*Chen et al., 2013*). A post-processing program in the software package Relion (*Scheres, 2012*) was used to calculate the FSC between two halves of the maps. The statistics of the structures are summarized in *Table 3* and the FSC plots are shown in the supplementary figures. The

**Table 1.** Estimated Molecular Weight of MIPs associated with Triple.

| Name | Estimated MW (KD) | Periodicity | Location |
|---|---|---|---|
| MIP1 | 45 | 8 nm | cone-like structure associated with the lumen side of A5 |
| MIP2 | n/a | 4 nm | trellis-like structure spanning laterally from A11 to A13 |
| MIP3 | 24 | 8 nm | laterally link A13, A1 and A2 in the lumen of A-tubule |
| MIP4 | 28 | 8 nm | laterally link A2 and A3 in the lumen of A-tubule |
| MIP5 | 74 | 8 nm | inner junction of A, B-tubule, laterally link A1, A2 and B10 |
| MIP6 | 27 | 8 nm | inner junction of A, B-tubule, laterally link A1, A13, B9, B10 |
| MIP7 | n/a | 8 nm | fin-like filamental structures running longitudinally along B5, B4 B3 in the lumen of B-tubule |
| MIP8 | 15 | 8 nm | outside of B-tubule but in the lumen of C-tubule, laterally link B6 and B7 |
| MIP9 | 92 | 8 nm | inner junction of B, C-tubule, link B8, B9 and C10 |
| MIP10 | 19 | 8 nm | inner junction of B, C-tubule, link B7 and C10 |
| MIP11 | 6 | 8 nm | laterally crosslink C9 and C10 in the lumen of C-tubule |

Protein density = 0.849 Dalton/$Å^3$

Yellow: MIPs associated with the A-tubule of triplet

Blue: MIPs associated with the B-tubule of triplet

Green: MIPs associated with the C-tubule of triplet

DOI: https://doi.org/10.7554/eLife.43434.040

**Table 2.** Quantive Analysis on the Twist Motion of the A-C Linker.

**A. Measurement of Twist Angle at A-C Linker**

| Class | $\theta$ angle (degree) |
|---|---|
| I | 93 |
| II | 90 |
| III | 87 |
| IV | 83 |

**B. Measurement of Twist Angle at A-C Linker**

| Class | $\theta$ angle (degree) |
|---|---|
| I | 96 |
| II | 93 |
| III | 90 |
| IV | 89 |
| V | 84 |
| VI | 80 |

A. The assignment for each class is following convention defined in **Figure 4D** and it is based on the resulting four classes in the *first* classification scheme, illustrated in **Figure 3—figure supplement 1**. In the scheme, the A-tubule is used as a reference point. The angle θ is defined as the angle between the Arm A and the Leg B as illustrated in **Figure 4A**.

B, The assignment for each class is following convention defined in **Figure 4E** and it is based on the resulting six classes in the *second* classification scheme, illustrated in **Figure 3—figure supplement 4**. In the scheme, the C-tubule is used as a reference point.

In both A and B, the angle θ is defined as the angle between the Arm A and the Leg B as illustrated in Figure 4A

DOI: https://doi.org/10.7554/eLife.43434.030

local resolution of the pinhead structure, as shown in **Figure 2—figure supplement 14**, is estimated by *blocres* in Bsoft (**Heymann and Belnap, 2007**).

UCSF Chimera was used for displaying the surface-rendered averaged structures, the model building and the volume segmentation (**Goddard et al., 2005**).

To segment out the non-tubulin components, the MIPS, bound to the triplet and to estimate their molecular weights, an atomic model of MT protofilament was docked into the subtomogram averages of the A-, B- and C-tubules, respectively. These resulted in the atomic models for all three tubules (**Figure 2**). The differences between the subtomogram average map and the model density were attributed to the MIPs. The MIP density was segmented in UCSF Chimera. Based on the volume of segmented MIPs and assuming the protein density is 0.849 Dalton/$Å^3$ (**Fischer et al., 2004**), the approximate molecular weight of each MIP was estimated (**Table 1**).

To model Poc1 in the A-C linker structure, a crystal structure of WD40 β-propeller domain with seven repeats (PDB ID 1S4U) was first manually docked into the doughnut-shaped density in Leg B of the A-C linker. The docking was further optimized by using the 'Fit in Map' function in UCSF Chimera. A local cross correlation coefficient between the model and the density map is 0.80, indicating a good fit.

## Subtomogram classification and alignment on the A-C linker

For classification on the A-C linker structure, based on the initial alignment result of the triplet and by using the A- or C-tubule as reference point, respectively, we searched for subtomograms that exhibit intact A-C linker. In two parallel schemes, we identified 3992 subtomograms from 4 classes and 3245 subtomogram from five classes (Class I with 233 subtomograms was excluded due to its small number and low SNR) that show relatively intact A-C linker attached to the A- or the C-tubule, respectively (**Figure 3—figure supplements 1,4**). These two datasets are subjected to further refinement. The results are two structures, both at 23.1 Å resolution, showing more complete and detailed structure of the A-C linker (**Figure 3—figure supplements 2,3,5**).

To generate a complete structure of the A-C linker, since we have identified multiple classes of subtomogram showing intact A-C linker (**Figure 3—figure supplements 1,4**), though each in relative low resolution, we docked the above two averages (**Figure 3—figure supplements 2,5**) into these

low-resolution class-averages. The result is a structure having complete A-C linker for each identified class. An example process is illustrated in *Figure 3—figure supplement 6*.

## Analysis on the periodicity of the pinhead structure

For the pinhead structure, we carried out extensive studies on its longitudinal periodicity. There are several factors that can potentially affect the periodicity in the average. These are, 1) the length of the triplet in subtomogram, 2) the percentage of overlap between neighboring triplet segments, 3) the structural component included in the soft-edge mask that is used in subtomogram alignment, 4) a reference is provided or not during the initial alignment, 5) the structure difference between the triplet in the procentriole and the triplet from the proximal region of the mother centriole. All these factors can potentially introduce errors in the subtomogram alignment. We took these factors into account and did extensive tests. All results consistently show 8 nm periodicity in the pinhead structure. In the final map of the pinhead, the longitudinal length is 19 nm, including more than two repeats of the pinhead.

## Acknowledgement

We thank Michael Braunfeld and Shawn Zheng for advice on tomography data collection, Matt Harrington for supporting the computational infrastructure, Hiroaki Ishikawa for discussion on the biology of *Chlamydomonas*, centrioles and basal bodies, Tom Goddart for help with UCSF Chimera program. We are grateful to many of our colleagues for critical reading of the manuscript and for the encouragements. This work is supported in part by NIH grants GM031627 (DAA), GM118099 (DAA), PO1 GM105537 (DAA), GM113602 (WFM) and by HHMI (DAA) and by the Spanish AEI/FEDER (SAF2017-84565-R) (J-JF) and by Fundacion Ramon Areces (J-JF).

## Additional information

### Funding

| Funder | Grant reference number | Author |
|---|---|---|
| Howard Hughes Medical Institute | | David A Agard |
| National Institutes of Health | GM031627 | David A Agard |
| National Institutes of Health | GM118099 | David A Agard |
| National Institutes of Health | PO1 GM105537 | David A Agard |
| National Institutes of Health | GM113602 | Wallace F Marshall |
| Fundación Ramón Areces | | Jose-Jesus Fernandez |
| Spanish AEI/FEDER | SAF2017-84565-R | Jose-Jesus Fernandez |

The funders had no role in study design, data collection and interpretation, or the decision to submit the work for publication.

### Author contributions

Sam Li, Conceptualization, Data curation, Formal analysis, Validation, Investigation, Methodology, Writing—original draft, Writing—review and editing, conceived the project, prepared the sample, collected the data, developed the algorithms and methods, processed, analyzed and interpreted the data; Jose-Jesus Fernandez, Data curation, Software, Formal analysis, Validation, Investigation, Methodology, Writing—review and editing, developed the algorithms and methods, processed, analyzed and interpreted the data; Wallace F Marshall, Resources, Funding acquisition, Writing—review and editing, secured the funding; David A Agard, Resources, Supervision, Funding acquisition, Project administration, Writing—review and editing, secured the funding, supervised the project

## Author ORCIDs

Sam Li http://orcid.org/0000-0002-0210-8192
Jose-Jesus Fernandez https://orcid.org/0000-0003-2222-3355
Wallace F Marshall http://orcid.org/0000-0002-8467-5763
David A Agard https://orcid.org/0000-0003-3512-695X

## Decision letter and Author response

Decision letter https://doi.org/10.7554/eLife.43434.059
Author response https://doi.org/10.7554/eLife.43434.060

# Additional files

## Supplementary files

• Transparent reporting form
DOI: https://doi.org/10.7554/eLife.43434.041

## Data availability

8 structures based on the subtomogram averaging have been deposited in the EMDB under the accession codes: EMD-9167, EMD-9168, EMD-9169, EMD-9170, EMD-9171, EMD-9172, EMD-9173, EMD-9174

The following datasets were generated:

| Author(s) | Year | Dataset title | Dataset URL | Database and Identifier |
|---|---|---|---|---|
| Li S, Fernandez JJ, Marshall WF, Agard DA | 2019 | Electron cryo-tomography and subtomogram averaging of microtubule triplet from procentriole | http://www.ebi.ac.uk/pdbe/entry/emdb/EMD-9167 | Electron Microscopy Data Bank, EMD-9167 |
| Li S, Fernandez JJ, Marshall WF, Agard DA | 2019 | Electron cryo-tomography and subtomogram averaging of microtubule triplet from procentriole | http://www.ebi.ac.uk/pdbe/entry/emdb/EMD-9168 | Electron Microscopy Data Bank, EMD-9168 |
| Li S, Fernandez JJ, Marshall WF, Agard DA | 2019 | Electron cryo-tomography and subtomogram averaging of microtubule triplet from procentriole | http://www.ebi.ac.uk/pdbe/entry/emdb/EMD-9169 | Electron Microscopy Data Bank, EMD-9169 |
| Li S, Fernandez JJ, Marshall WF, Agard DA | 2019 | Electron cryo-tomography and subtomogram averaging of microtubule triplet from procentriole | http://www.ebi.ac.uk/pdbe/entry/emdb/EMD-9170 | Electron Microscopy Data Bank, EMD-9170 |
| Li S, Fernandez JJ, Marshall WF, Agard DA | 2019 | Electron cryo-tomography and subtomogram averaging of microtubule triplet from procentriole | http://www.ebi.ac.uk/pdbe/entry/emdb/EMD-9171 | Electron Microscopy Data Bank, EMD-9171 |
| Li S, Fernandez JJ, Marshall WF | 2019 | Electron cryo-tomography and subtomogram averaging of microtubule triplet from procentriole | http://www.ebi.ac.uk/pdbe/entry/emdb/EMD-9172 | Electron Microscopy Data Bank, EMD-9172 |
| Li S, Fernandez JJ, Marshall WF, Agard DA | 2019 | Electron cryo-tomography and subtomogram averaging of microtubule triplet from procentriole | http://www.ebi.ac.uk/pdbe/entry/emdb/EMD-9173 | Electron Microscopy Data Bank, EMD-9173 |
| Li S, Fernandez JJ, Marshall WF, Agard DA | 2019 | Electron cryo-tomography and subtomogram averaging of microtubule triplet from procentriole | http://www.ebi.ac.uk/pdbe/entry/emdb/EMD-9174 | Electron Microscopy Data Bank, EMD-9174 |

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
