## [Decision Letter]

Thank you for submitting your article "Electron Cryo-Tomography Provides Insight into Procentriole Architecture and Assembly Mechanism" for consideration by *eLife*. Your article has been reviewed by Anna Akhmanova as the Senior Editor, Sriram Subramaniam as the Reviewing Editor and three reviewers. The following individual involved in review of your submission has agreed to reveal his identity: Masahide Kikkawa (Reviewer #1).

The reviewers have discussed the reviews with one another and the Reviewing Editor has drafted this decision to help you prepare a revised submission.

Summary:

All three reviewers agreed that this manuscript provides important new information on the structure of the procentriole microtubule and assembly intermediates using cryo-electron tomography. The use of subtomogram classification and averaging to reveal conformational changes within the *C. reinhardtii* centrioles was also seen as a valuable contribution. However, all reviewers had several significant concerns that need to be addressed in order for the manuscript to be acceptable for publication in *eLife*.

Essential revisions:

1) One major issue is POC1. In subsection “The Twist of A-C linker Results in Iris Diaphragm Motion of the Procentriole” and Figure 3, the authors state that POC1 is an integral component of the A-C linker. The reviewers believe this is an overinterpretation. To claim that "combined with the prevailing data, our structure provides evidence that Poc1 is an integral component of the procentriole A-C linker that bridges neighboring triplets." (Subsection “The Twist of A-C linker Results in Iris Diaphragm Motion of the Procentriole”), the authors would have to identify the 3D position of POC1 by analyzing POC1 mutant or structural labeling of POC1 using cryo-electron tomography. Alternatively, the author could weaken the statement.

POC1 localization to the leg B of the A-C linker is inferred based on previous work and its predicted 3D structure. Yet, POC1 has not been definitively identified as a component of linker. Further, POC1 does not appear to be localized only to the proximal ends of the basal bodies in *Chlamydomonas*, as stated. Keller at al. (2009) showed that POC1 also localizes to both inner and outer centriole walls and is present on the entire length of the centriole. Until there is direct evidence for the localization of POC1 to the linker, the conclusion needs to be treated as hypothetical, especially in the Abstract. The Abstract thus should be revised. Furthermore, the reviewers recommend that the authors review the procentriole proteome and itemize possible β-propeller candidates as a supplementary table.

2) Figure 1C represents the subtomogram average of MT triplets from mother centriole proximal ends and procentrioles. The reviewers would like to see explicit clarification for combining these tomograms, especially given the fact that Figure 1—figure supplement 2 demonstrates structural differences between the proximal end of the mother centriole and procentriole.

3) One reviewer asks whether the density identified between C10 and B8 protofilaments (MIP9 in the B-C inner junction) could correspond to the eleventh protofilament, taking into account the data in Figure 1D, the density maps in Figure 2—figure supplement 9, and especially the procentriole model in Figure 1C. Is there evidence to rule out the possibility that MIP9 represents a protofilament with some additional associated material? In Figure 1C this density appears indistinguishable from other protofilament densities and it has similar periodicity as tubulin in the density map in Figure 2—figure supplement 9. If the possibility cannot be ruled out, it should be discussed.

4) The authors propose that the A-C linker in procentrioles during C-tubule assembly is "visible albeit with weaker density due to its flexible nature". How do they argue against the possibility that it is not visible because it still does not contain all components at this point in procentriole assembly?

5) The authors state that they "identified two triplet intermediates" and "proposed the model for procentriole triplet assembly". However, Anderson and Brenner (1971) and Guichard et al. (2010) already proposed the model for lateral tubulin addition to the microtubules. Anderson even provides some evidence for partially assembles microtubules. MT intermediates are here better documented, but the original work must be credited.

6) Figure 1B does not show two well preserved procentrioles attached to the mother centriole as stated in Results. It is not evident form Figure 1B that the procentriole is attached to the mother centriole. Further what 'attached' means in structural terms is unclear. Procentrioles in Video 1 are adjacent but clearly not attached (See O'Toole and Dutcher work describing basal body duplication cycle and organization).

7) The Introduction describes the centriole cycle as it appears in vertebrate cells. However, the authors do not offer any introduction about basal body duplication in *Chlamydomonas*. That is indeed perplexing given significant differences in the duplication process and timing between the two organisms. Very limited description of *Chlamydomonas* basal body complexes can be found in 'Overall architecture of the procentriole' but there is no reference to address readers to ample published work describing ultrastructural features of *Chlamydomonas* basal body complexes. A scheme or a classical TEM illustration of an intact basal body complex would be helpful as well.

8) One reviewer notes that there is poor citation of prior work on procentriole structure, procentriole formation, and MT linkers, please address this.

9) Two reviewers seek explicit discussion of why exactly the authors think that MIP9 is not a tubulin protofilament.

---

## [Author Response]

Essential revisions:1) One major issue is POC1. In subsection “The Twist of A-C linker Results in Iris Diaphragm Motion of the Procentriole” and Figure 3, the authors state that POC1 is an integral component of the A-C linker. The reviewers believe this is an overinterpretation. To claim that "combined with the prevailing data, our structure provides evidence that Poc1 is an integral component of the procentriole A-C linker that bridges neighboring triplets." (Subsection “The Twist of A-C linker Results in Iris Diaphragm Motion of the Procentriole”), the authors would have to identify the 3D position of POC1 by analyzing POC1 mutant or structural labeling of POC1 using cryo-electron tomography. Alternatively, the author could weaken the statement.POC1 localization to the leg B of the A-C linker is inferred based on previous work and its predicted 3D structure. Yet, POC1 has not been definitively identified as a component of linker. Further, POC1 does not appear to be localized only to the proximal ends of the basal bodies in Chlamydomonas, as stated. Keller at al. (2009) showed that POC1 also localizes to both inner and outer centriole walls and is present on the entire length of the centriole. Until there is direct evidence for the localization of POC1 to the linker, the conclusion needs to be treated as hypothetical, especially in the Abstract. The Abstract thus should be revised. Furthermore, the reviewers recommend that the authors review the procentriole proteome and itemize possible β-propeller candidates as a supplementary table.

We completely agree with the reviewers’ comments that our structure, along with other previous studies can only infer that POC1 might be an A-C linker component in the procentriole. More experiments and analysis are needed for conclusive localization of the POC1. Following the reviewers’ suggestion, we have changed our manuscript in several locations –

1) In the Abstract, we have changed it to “The structure *infers* that POC1 is *likely* an integral component of A-C linker.”

2) In subsection “Structure of the A-C linker that Connects Neighboring Triplets”, now we state that –

“Thus, combined with these data, our structure suggests that Poc1 is *likely* an integral component of the procentriole A-C linker that bridges neighboring triplets. Its N-terminal WD40 β-propeller domain provides multiple sites for interacting with other A-C linker components. However, in addition to procentriole, Poc1 has also been found at other locations in mother centriole and in flagellum (Keller et al., 2009; Pearson et al., 2009). A precise localization of POC1 gene product on the centriole has to wait for future higher resolution structure by cryoET and other studies.”

3) In the figure legend for Figure 3 (D and E), now we have changed to “Close up views of the Leg B in the A-C linker show a *possible* location of Poc1.”

Previous study on the *Chlamydomonas* basal body proteome has identified at least three major centriole/cilia components that contain predicted WD-40 repeats. These are POC1, POC16 and BUG14 (also known as FAP52) (Keller et al., 2009). Recent studies in *Chlamydomonas* have shown that POC16 is likely localized in the central core region of the centriole (Hamel et al., 2017), while FAP52 is a flagellar axoneme component localized to the lumen of the B-tubule (Owa et al., We also agree with reviewer that a thorough re-analysis on the entire centriole proteome and itemizing possible β-propeller or perhaps other motifs/domains candidates will be useful to localize components in the 3D structure, however it will be beyond the scope of this study.

2) Figure 1C represents the subtomogram average of MT triplets from mother centriole proximal ends and procentrioles. The reviewers would like to see explicit clarification for combining these tomograms, especially given the fact that Figure 1—figure supplement 2 demonstrates structural differences between the proximal end of the mother centriole and procentriole.

Figure 1C shows the MT triplets structure based on the average of 12937 subtomograms. These are from two fractions of the data, the proximal ends of mother centriole (10854 subtomograms from 201 mother centrioles) and procentriole (2083 subtomograms from 110 procentrioles). In order to find possible structure difference between them, we averaged these two fractions separately (Figure 1—figure supplement 2). Comparison of these two averages reveals notable differences that are mainly localized on the B- and C-tubules, as indicated in Figure 1—figure supplement 2. As we have stated in the manuscript (subsection “Overall Architecture of the Procentriole”), we believe these differences are likely due to either transient association of proteins or the defective or incomplete structures, these include the intermediate structures of the B- and C-tubule that we identified and reported in the manuscript.

We’ve realized that our original statement may cause confusion. Now in subsection “Overall Architecture of the Procentriole” we have revised our statement as: “These differences may reflect difference in kinetics or transient association of proteins at various stages of assembly, or incompletion of procentriole assembly (see below for further detailed analysis on the B- and C-tubules in the procentrioles).”

In addition, as mentioned in the manuscript, we have carried out extensive focused classification on 2083 subtomograms on the A, B and C tubules from 110 procentrioles. This allowed us to identify 420, 758 and 862 defective or incomplete structure in each tubule respectively. These defective/incomplete subtomograms were excluded from further focused alignment on each tubule, resulting in more homogeneous datasets. Compared to the average of the entire triplet, the improvements are evident in the increased resolutions shown in Figure 1—figure supplement 1 and in Table 1. These averaged tubule structures were used for further segmentation and estimation of the molecular weight of MIPs associated to the tubule.

Now we have added this detailed description on the Material and methods section.

3) One reviewer asks whether the density identified between C10 and B8 protofilaments (MIP9 in the B-C inner junction) could correspond to the eleventh protofilament, taking into account the data in Figure 1D, the density maps in Figure 2—figure supplement 9, and especially the procentriole model in Figure 1C. Is there evidence to rule out the possibility that MIP9 represents a protofilament with some additional associated material? In Figure 1C this density appears indistinguishable from other protofilament densities and it has similar periodicity as tubulin in the density map in Figure 2—figure supplement 9. If the possibility cannot be ruled out, it should be discussed.

Though the filament formed by MIP9 resembles a MT protofilament (pf) from the top view (Figure 1C, Figure 2C and Figure 2—figure supplement 9) and it exhibits 4 nm longitudinal periodicity, when docking a model of MT protofilament into the MIP9 density, it shows substantial deviation from a canonical MT protofilament. Therefore, we concluded this likely is not a canonical MT protofilament and we assigned it as MIP9 that crosslinks C10 to B8 and B9 at the inner junction of B- and C-tubules.

To make this clear, now we have added an additional supplementary figure as Figure 2—figure supplement 12, showing modeling a MT protofilament into the MIP9 filament density and the structure deviation from a MT protofilament is clear.

In addition, we add a statement in subsection “Non-tubulin Components Associated with the Procentriole Triplet” in the main text –

“Among them, the MIP9 at the inner B-C tubule junction, is of particular interest. Even though it forms a filament exhibiting 4 nm periodicity, the structure deviates from a canonical MT protofilament (Figure 2—figure supplement 12). Therefore, we assigned it as a non-tubulin protein MIP9.”

4) The authors propose that the A-C linker in procentrioles during C-tubule assembly is "visible albeit with weaker density due to its flexible nature". How do they argue against the possibility that it is not visible because it still does not contain all components at this point in procentriole assembly?We agree with the reviewer that it is possible that the A-C linker assembly might be incomplete at this point. Now we have modified our statement as “Interestingly, in all three averages, even though the C-tubule is incomplete, the A-C linker is visible albeit with weaker density likely due to its flexible nature or incomplete assembly.”5) The authors state that they "identified two triplet intermediates" and "proposed the model for procentriole triplet assembly". However, Anderson and Brenner (1971) and Guichard et al. (2010) already proposed the model for lateral tubulin addition to the microtubules. Anderson even provides some evidence for partially assembles microtubules. MT intermediates are here better documented, but the original work must be credited.

We thank the reviewer for pointing out this important reference. Now we have cited Anderson and Brenner’s work, along with Guichard et al. findings. In the Discussion section, we now state –

“Previous EM studies on mammalian centriole and basal body showed the triplet assembly was a non-uniform process where in each triplet the A-tubule assembled first, followed by sequential B- and C-tubule assembly. During the tubule elongation, their lengths varied substantially (Anderson and Brenner, 1971; Guichard et al., 2010). Furthermore, by analyzing the nascent procentrioles, Anderson and Brenner observed the B- and C-tubule first emerged as sheets laterally attached to the A- and B-tubule respectively. Consistent with these findings, by analyzing a population of procentrioles, we found the average length of three tubules in the triplet varies substantially. In addition, we have identified two triplet intermediates, showing that the B-tubule initiates assembly uni-directionally from the outer A-B junction.”

6) Figure 1B does not show two well preserved procentrioles attached to the mother centriole as stated in Results. It is not evident form Figure 1B that the procentriole is attached to the mother centriole. Further what 'attached' means in structural terms is unclear. Procentrioles in Video 1 are adjacent but clearly not attached (See O'Toole and Dutcher work describing basal body duplication cycle and organization).

Unlike the centriole in vertebrates where the procentriole is directly linked to the mother centriole, in *Chlamydomonas* the attachment of procentriole to the mother centriole is via a set of rootlet microtubules (rMT). This has been well documented and has been observed throughout our reconstructions of the nuclear-flagellar-apparatus (NFAp).

To make this clear, we have put a label of rootlet MT (rMT) in Figure 1B, indicating the likely attachment point of the procentriole and modified the main text (subsection “Overall Architecture of the Procentriole”) as “…. two well-preserved procentrioles are attached to their respective mother centrioles via the rMT (Figure 1B, Video 1).”

In addition, we have also modified the figure legend related to Figure 1B as “A slice from the reconstructed tomogram of procentriole attached to the mother centriole via the rMT.”

7) The Introduction describes the centriole cycle as it appears in vertebrate cells. However, the authors do not offer any introduction about basal body duplication in Chlamydomonas. That is indeed perplexing given significant differences in the duplication process and timing between the two organisms. Very limited description of Chlamydomonas basal body complexes can be found in 'Overall architecture of the procentriole' but there is no reference to address readers to ample published work describing ultrastructural features of Chlamydomonas basal body complexes. A scheme or a classical TEM illustration of an intact basal body complex would be helpful as well.

We thank reviewer for pointing this out. Indeed, there are significant differences in term of morphology, ultrastructure and timing of the centriole duplication between *Chlamydomonas* and the vertebrate cells. As the reviewer said, there exist a large amount of previously published studies documenting the centriole duplication and cell cycle in *Chlamydomonas.* Now we have adapted reviewer’s comment and introduce this point in the Introduction as well as cited several additional references. We now state –

“…Similar to the centriole duplication in metazoans such as in mammals, the centriole duplication process in the unicellular organism *Chlamydomonas reinhardtii* is tightly controlled. The process has been described in a series of seminal studies (Cavalier-Smith, 1974; Geimer and Melkonian, 2004; O'Toole and Dutcher, 2014). Compared to the vertebrates, despite many morphological and ultrastructural differences in the duplication steps, a number of key components in the *Chlamydomonas* centriole assembly have been found conserved in other higher eukaryotes (Dutcher et al., 2002; Dutcher and Trabuco, 1998; Hiraki et al., 2007; Keller et al., 2009; Matsuura et al., 2004; Nakazawa et al., 2007).”

8) One reviewer notes that there is poor citation of prior work on procentriole structure, procentriole formation, and MT linkers, please address this.

We have now cited more previous works on the procentriole structure, assembly and A-C linker. These include:

Cavalier-Smith, 1974

Geimer and Melkonian, 2004

O’Toole and Dutcher, 2014

Anderson and Brenner, 1971

Guichard et al., 2010

9) Two reviewers seek explicit discussion of why exactly the authors think that MIP9 is not a tubulin protofilament.

Please see our answer to the third point above.